# AutoLR: A Method for Automatic Tuning of Learning Rate

## Abstract

One very important hyperparameter for training deep neural networks is the learning rate of the optimizer. The choice of learning rate schedule determines the computational cost of getting close to a minima, how close you actually get to the minima, and most importantly the kind of local minima (wide/narrow) attained. The kind of minima attained has a significant impact on the generalization accuracy of the network. Current systems employ hand tuned learning rate schedules, which are painstakingly tuned for each network and dataset. Given that the state space of schedules is huge, finding a satisfactory learning rate schedule can be very time consuming. In this paper, we present *AutoLR*, a method for auto-tuning the learning rate as training proceeds. Our method works with any optimizer, and we demonstrate results on SGD, Momentum, and Adam optimizers.

We extensively evaluate *AutoLR* on multiple datasets, models, and across multiple optimizers. We compare favorably against state of the art learning rate schedules for the given dataset and models, including for ImageNet on Resnet-50, Cifar-10 on Resnet-18, and SQuAD fine-tuning on BERT. For example, *AutoLR* achieves an EM score of 81.2 on SQuAD v1.1 with $BERT_{BASE}$ compared to 80.8 reported in (Devlin et al. (2018)) by just auto-tuning the learning rate schedule. To the best of our knowledge, this is the first automatic learning rate tuning scheme to achieve state of the art generalization accuracy on these datasets with the given models.

## 1 Introduction

Learning rate is one of the most important hyperparameters that impact deep neural network (DNN) training performance. It determines how fast we get close to a minima, which in turn determines the computational cost of optimization. It also determines how close we get to the minima, e.g. higher learning rates may get in the neighborhood of a minima much faster, but then just bounce at a height above the minima (Xing et al. (2018); Nar & Sastry (2018)). The learning rate also determines the kind of minima (e.g. wide vs narrow) attained (Keskar et al. (2016)), which has a significant impact on the generalization accuracy.

Therefore, it is not surprising that a lot of effort has gone into automatically tuning the learning rate (Schraudolph (1999); Schaul et al. (2013b); Rolinek & Martius (2018)). However, till date, none of these techniques have been able to deliver state of the art test accuracy on standard benchmarks. Instead, deep learning researchers today rely on a mixture of brute force search, augmented with simple heuristics such as using a staircase, polynomial, or exponential decay-based learning rate schedules. This problem is further exacerbated by the fact that different optimizers such as SGD or Adam work best on different datasets and require very different learning rate schedules.

For example, while training image datasets such as Cifar-10 and ImageNet with SGD or Momentum, a staircase schedule with high starting learning rate typically performs best. On the other hand, with the Adam optimizer, a smaller starting learning rate is generally used. Further, in NLP tasks such as machine translation with Adam, typically a linear warmup is followed by polynomial decay, e.g. the Transformer network (Vaswani et al. (2017)) uses a linear warmup followed by an inverse square root decay. Similar schedules are used for BERT (Devlin et al. (2018)) pre-training, while BERT fine-tuning uses a linear decay.

Our key idea to tackle this problem is driven by the following simple observation. Consider training Cifar-10/ImageNet using the typical staircase schedule. One can observe that the initial high learning

rate results in training loss stagnating after a few epochs. However, if one were to decay the learning rate when loss stagnates, the achieved test accuracy is considerably lower as compared to letting the high learning rate continue for longer, despite no apparent improvement in training loss.

Our insight is that this initial phase of training using a high learning rate, even with no improvement in loss, is crucial for generalization and is one of the key missing pieces from prior attempts at automatic learning rate tuning. We hypothesize that for DNNs, the number of narrow minimas far outnumber the wide minimas. To generalize well, we want the optimizer to land in wide minimas. An interesting intuitive observation is that a large learning rate can escape narrow minimas easily (as the optimizer can jump out of them with large steps), however once it reaches a wide minima, it is likely to get stuck in it (if the "width" of the wide minima is large compared to the step size).

The above hypothesis motivates our *Explore-Exploit* scheme where we force the optimizer to first *explore* the landscape with a high learning rate for sometime in order to land in a wide minima. We should give the *explore* phase enough time so that the probability of landing in a wide minima is high. Once we are confident that the optimizer is stuck in the vicinity of a wide minima, we activate the *Exploit* phase of *AutoLR*.

The basic idea of the *exploit* phase is as follows. We first look at how local perturbations in the current learning rate impact the training loss. Since we only look at local perturbations, we can model the loss as a function of these perturbations via a Taylor series expansion. We then make a quadratic approximation, and solve for the optimal perturbation in the learning rate which will minimize the loss. This is similar to Newton's method, but applied only in the descent direction (section 2). We do extensive validation that the quadratic is a good approximation here. Although the basic idea is simple, it is complicated by the fact that deep learning relies on stochastic optimization methods, such as SGD, which causes the loss values to be noisy across mini-batches, potentially throwing off our estimates. We discuss how we handle stochasticity in section 2.2.

We demonstrate the efficacy of the explore-exploit approach by evaluating *AutoLR* across a wide range of models and datasets, ranging from NLP (SQuAD on BERT-base, Transformer on IWSLT) to CNNs (e.g. ImageNet on ResNet-50, Cifar-10 on ResNet18), and across multiple optimizers: SGD, Momentum and Adam. In all cases, *AutoLR* matches or beats test accuracy of state-of-the-art hand-tuned learning rate schedules. For example, on SQuAD v1.1 fine-tuning with BERT$_{\text{BASE}}$, *AutoLR* is able achieve an EM score of 81.2, compared to 80.8 reported in Devlin et al. (2018) by just auto-tuning the learning rate schedule (all other parameters were unchanged). To the best of our knowledge, *AutoLR* is the first automatic learning rate scheduler to achieve state of the art results on these datasets for the given models. We show extensive evaluation of our method in section 3.

The main contributions of our work are:

1. The observation that an initial Explore phase with high learning rate is crucial for good generalization in automatic learning rate schemes.
2. Incorporating this observation via an *Explore-Exploit* approach with a novel exploitation scheme for tuning learning rate using a local approximation of the optimization landscape.
3. The first automatic learning rate tuning scheme that beats or achieves generalization of state of the art learning rate schedules in multiple models/datasets including ImageNet.

## 2 METHOD

Let $L(\theta)$ be the loss of the network as a function of its parameters $\theta$. Note that in practice, since we work with stochastic optimizers such as SGD, the loss computed in each minibatch is only an approximation of the true loss. This distinction is important when we come to the estimation of the best learning rate, but can be ignored for the formulation below.

The loss of the network in the next time step is $L(\theta - \eta\vec{d})$, where $\vec{d}$ is the step direction, and $\eta$ is the learning rate. If we think of $L(\theta - \eta\vec{d})$ as a function of the learning rate $\eta$, our task is to find an $\eta$ which minimizes this function. Since it is hard to directly estimate $L$ as a function of $\eta$, we instead consider what happens if we perturb $\eta$ by a small amount $\epsilon$, i.e we look at $L(\theta - (\eta + \epsilon)\vec{d})$. Looking

at this as a function of $\epsilon$, and applying Taylor series expansion we get,

$$\hat{L}(\epsilon) = L(\theta - (\eta + \epsilon)\vec{d}) = L(\theta - \eta\vec{d} - \epsilon\vec{d})$$
$$= L(\theta - \eta\vec{d}) - \epsilon\vec{d}^T\vec{g} + \frac{1}{2}\epsilon^2\vec{d}^T H\vec{d} + \mathcal{O}(\epsilon^3), \tag{1}$$

where $\vec{g}$ is the gradient of $L$ w.r.t $\theta$, and $H$ is the Hessian. Note that computing accurate value of the gradient $\vec{g}$ is expensive, as it requires going over the entire dataset (not just a minibatch), while computing the Hessian $H$ is typically intractable for deep networks which have millions of parameters. However, it turns out that we don't actually need the values of $\vec{g}$ and $H$. In fact, we can just look at $\hat{L}$ as a quadratic function of $\epsilon$, i.e.

$$\hat{L}(\epsilon) = k_0 + k_1\epsilon + k_2\epsilon^2 + \mathcal{O}(\epsilon^3). \tag{2}$$

To compute $\{k_0, k_1, k_2\}$, we simply evaluate the loss at a few values of $\epsilon$ and fit a quadratic polynomial. Note that we are able to get away with having to calculate the high dimensional $\vec{g}$, and $H$, because the loss function only looks at the effect of these first and second order derivatives in *one single direction* $\vec{d}$, as determined by the terms $\vec{d}^T\vec{g}$ and $\vec{d}^T H\vec{d}$. Once we know the values of $\{k_0, k_1, k_2\}$, we can simply minimize this quadratic to get the optimal value of $\epsilon_{min} = -\frac{k_1}{2k_2}$ to minimize the loss $L$ in the next time step. Also observe that our method only needs the search direction $\vec{d}$ for computing the various loss samples. As a result, our method works for any optimizer, as long as we can access the search direction. We have implemented our method for multiple optimizers including SGD, Momentum, and Adam.

Note that it is very important to use a perturbation $\epsilon$ for the Taylor expansion in equation 1, rather than just expanding on $\eta$ via $L(\theta - \eta\vec{d}) = L(\theta) - \eta\vec{d}^T\vec{g} + \frac{1}{2}\eta^2\vec{d}^T H\vec{d} + \mathcal{O}(\eta^3)$. This is because $\eta$ may not be small, causing the $\mathcal{O}(\eta^3)$ error to be quite large. See appendix A for empirical validation that a second order approximation captures the loss perturbations accurately.

**Epsilon Thresholding.** The quadratic approximation in equation 2 yields an optimal value of $\epsilon_{min} = -\frac{k_1}{2k_2}$ for the minimum loss $L$ at the next time step. Note that the quadratic approximation is valid only up to an $\mathcal{O}(\epsilon^3)$ error, while the minimization can yield $\epsilon_{min}$ with large absolute values, thereby making huge errors. That is, although our quadratic approximation is valid only for small values of $|\epsilon|$, the quadratic's minima may be beyond this approximation range, as illustrated in Figure 1. Thus, we need to threshold $\epsilon_{min}$ to a reasonable value. Since we are interested in approximating the loss to some precision, we use a relative threshold as follows. We approximate $\mathcal{O}(\epsilon^3)$ as $|\epsilon|^3$, and bound the error:

$$|\epsilon|^3 < r * L(\theta), \tag{3}$$

where $r$ is the relative error threshold.

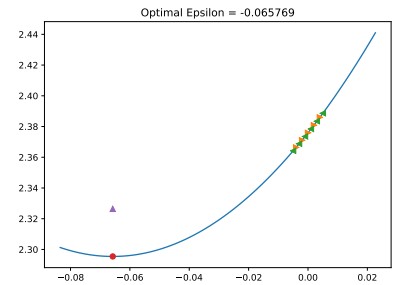

Figure 1: A large $|\epsilon_{min}|$ can give bad approximation. Orange and green triangles show loss samples used in fitting and testing respectively. Red circle shows the minimum loss value of the quadratic, and purple triangle shows the true loss there.

### 2.1 TACKLING GENERALIZATION

The choice of learning rate is an important factor in determining generalization quality of the trained network. The above method for tuning the learning rate is a local method, and does not take into account the highly non-linear and non-convex global optimization landscape of deep networks. As a result it can take globally suboptimal decisions which can impact generalization. In this section we discuss the problem in more detail, and our solution.

Although understanding generalization of deep neural networks is an open problem, there have been interesting findings recently. Kawaguchi (2016) found that deep neural networks have many local minimas, but all local minimas are also the global minima (also see Goodfellow et al. (2016)). Also, it is widely believed that wide minimas generalize much better than narrow minimas (Hochreiter & Schmidhuber (1997); Keskar et al. (2016); Jastrzebski et al. (2017); Wang et al. (2018)), even though

they have the same training loss. Keskar et al. (2016) found that small batch SGD generalizes better than large batch SGD and also lands in wider minimas, suggesting that noise in SGD acts as an implicit regularizer. Interestingly however, more recent work was been able to generalize quite well even with very large batch sizes (Goyal et al. (2017); McCandlish et al. (2018); Shallue et al. (2018)), by scaling the learning rate linearly as a function of the batch size. This suggests that the lack of noise in large batch SGD can be compensated with high learning rates, and thus the learning rate plays a crucial role in generalization.

Many popular learning rate schedules start the training with high learning rates, and then reduce the learning rate after every few epochs or following some hand tuned fall curve. An interesting observation in many such examples is that even though a high learning rate stagnates after a few epochs, one still needs to run enough iterations at the higher learning rate in order to get good generalization. For example, see Figure 2, which shows the training loss of a Resnet-18 model training on Cifar10 dataset at a fixed LR of 0.1 with SGD. The training loss stagnates after $\approx 50$ epochs, however as shown in Table 1[1], generalization continues to improve if we increase the number of epochs trained at a higher learning rate.

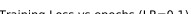

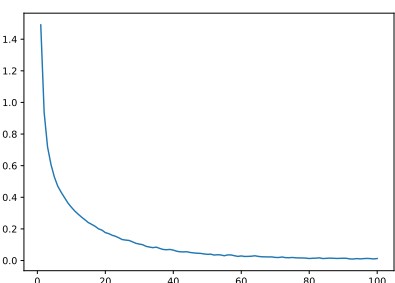

Figure 2: SGD training loss for Cifar-10 at LR of 0.1. Loss stagnates after $\approx 50$ epochs.

| Explore Epochs | Test Accuracy |
|---|---|
| 40 | 94.52 |
| 50 | 94.63 |
| 60 | 94.67 |
| 80 | 94.72 |

Table 1: Cifar-10 on Resnet-18 is trained for 200 epochs with Momentum. A LR of 0.1 is used for the explore epochs. Half the remaining epochs are trained at 0.01 and the other half at 0.001. Reported results are average over 4 runs.

To understand the above phenomena better, we run the following experiment. We train Cifar-10 on Resnet-18 using a high learning rate of 0.1 for only 40 epochs and then use learning rates of $10^{-2}$ and $10^{-3}$ for 80 epochs each. We do this training many times with different random initializations. On an average this yields a low generalization accuracy of 94.58 over 20 runs. However, we find that in 1 of the 20 runs our test accuracy reaches 94.98!

We compute the curvature of the loss surface at the end of training for the various runs and are able to validate that the high test accuracy indeed corresponds to a wider minima than the low test accuracy runs. Specifically, we use the highest eigenvalue[2] of the Hessian at the minima as a measure of the minima width (Jastrzebski et al. (2019); Keskar et al. (2016)). We find that the high accuracy runs consistently have smaller eigenvalues compared to low accuracy runs. For example the run with highest test accuracy of 94.98 (training loss: $1.52e - 3$) had an eigenvalue of 0.02, while the run with median accuracy of 94.58 (training loss: $1.56e - 3$) had an eigenvalue of 0.05, and the run with minimum accuracy of 94.32 (training loss: $1.5e - 3$) had an eigenvalue of 0.10.

**Hypothesis: low density of wide minima in the loss landscape of deep neural networks.** To explain the observation that *training at high learning rate for long duration consistently achieves high test accuracy while training at high learning rate for shorter duration until train loss saturates rarely achieves high test accuracy*, we hypothesize that for deep neural networks, narrow minima far outnumber wide minima. An intuitive explanation of the observed phenomena is that a large learning rate can escape narrow minima "valleys" easily (as the optimizer can jump out of them with large steps), however once it reaches a wide minima "valley", it is likely to get stuck in it (if

---

[1]We found a bug in our previous Cifar-10 Resnet model, where while computing the test accuracy we used the test batch's mean/variance for batch normalization layers, rather than using the running average from training. We have fixed this bug and reran all Cifar experiments. The numbers here and elsewhere for Cifar experiments have been updated with this fix.

[2]We used the opensource implementation at `https://github.com/noahgolmant/pytorch-hessian-eigenthings`

the "width" of the wide valley is large compared to the step size). See Jastrzebski et al. (2019) and Wu et al. (2018) for a discussion on relationship between learning rate and minima shape. When the density of wide minima is low, using a high learning rate for long enough duration should still be able to find the wide minima eventually. However, using a high learning for a short duration will only find the wide minima with some probability based on the chance of landing in the wide minima early during training. Since we find empirically that the probability of landing in the wide minima early in training is low (about 1/20 based on our experiments), we hypothesize that wider minima are fewer in number than narrow minima. Please see Section 4 for another experiment in the literature that adds more evidence to this hypothesis.

**Explore-Exploit:** The above hypothesis motivates our *Explore-Exploit* scheme where we force the optimizer to first *explore* the landscape with a high learning rate for sometime in order to land in the vicinity of a wide minima. During the explore phase, we only allow a monotonic increase of LR and disallow any decrease even if suggested by the local quadratic fit. We should give the *explore* phase enough time so that the probability of landing in a wide minima is high. Since the ratio of number of narrow vs wide minimas can depend on the architecture, dataset, etc., predicting the right explore phase duration is hard, and is currently a hyperparameter for *AutoLR*. Once we are confident that the optimizer must be now stuck in the vicinity of a wide minima, we start the *exploit* phase and allow *AutoLR* to reduce the learning rate as dictated by the local method to make fast progress and get close to the minima of this hopefully *wide* minima.

*Explore-Exploit* improves the generalization of *AutoLR* significantly. Without an explore phase, the *AutoLR* scheme will start reducing the learning rate as soon as it reaches close to a local minima, irrespective of whether it is a narrow or wide minima. As shown in Figure 3, locally at epoch 50, it makes sense to reduce the LR as it increases the rate of descent. Without any *explore* phase, *AutoLR* ( Eq 2), will make such a decision as well, potentially hurting generalization.

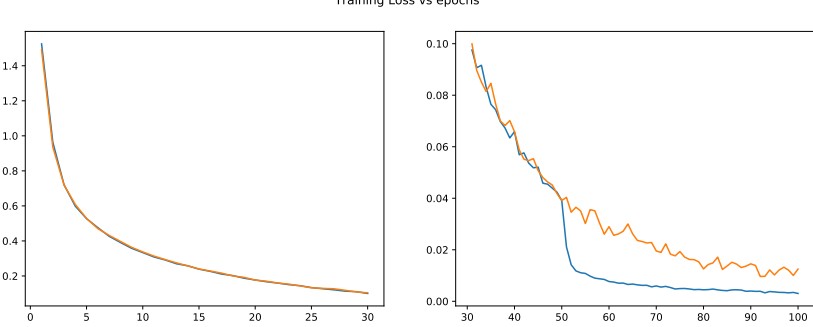

Figure 3: SGD training loss for Cifar-10 on Resnet-18. Plots are split across two to permit higher y-axis fidelity. In the orange plot, a fixed LR of 0.1 is used, while in the blue plot the LR is reduced from 0.1 to 0.01 at epoch 50. Clearly, at epoch 50, it locally makes sense to reduce the LR as it increases the rate of descent significantly.

| Explore Epochs | Test Accuracy |
|---|---|
| 0 | 93.77 |
| 10 | 94.01 |
| 25 | 94.27 |
| 50 | 94.52 |
| 70 | 94.58 |
| 80 | 94.62 |
| 100 | 94.79 |

Table 2: Cifar-10 on Resnet-18 trained for 200 epochs with Momentum. An LR of 0.1 is used for the explore epochs, and is determined by *AutoLR* for rest of the epochs. We report averages over 4 runs.

To validate the effectiveness of the explore-exploit scheme, we first analyze a state-of-the-art learning rate schedule for Cifar-10 on Resnet-18. This learning rate schedule trains with learning rates of $10^{-1}, 10^{-2}, 10^{-3}$ for 100, 50, 50 epochs each. The first 100 epochs at a high learning rate of 0.1, can be thought of as an explore phase here, as discussed in Figure 2. We experimented with reducing

this "explore" phase, and as shown in Table 1, it is clear that explore phase plays an important role in the final generalization accuracy. We now evaluate explore-exploit on the same model and dataset but with learning rate determined by *AutoLR* in the *exploit* phase, and try different duration of the *explore* phase. Table 2 shows the test accuracy for different number of explore epochs. As shown, the explore phase helps with improving generalization accuracy significantly from $93.77$ without explore to $94.79$ with 100 explore epochs.

**Saturation Threshold:** The above discussion suggests that overall, higher learning rates are better for generalization and thus one should prefer them even though a local analysis may suggest otherwise. To incorporate this idea into our method we employ what we call a *saturation threshold*. The idea is that we want to continue with the current learning rate and not lower it, unless it has *saturated* in terms of loss drop per iteration. That is, if the training loss drop rate has dropped below a threshold for the current learning rate, it suggests that the current learning rate has served its purpose, and we can move on to a lower learning rate if suggested so by *AutoLR*. We can either use an absolute threshold, or a relative threshold where we use the ratio of loss-drop rate when we started using the current learning rate to the current loss-drop rate. Finding the best absolute threshold for each model/dataset can be tricky, so we use a relative threshold, which we found easy to tune. Also, we noticed that the loss drop rate doesn't change drastically towards the end of the training when the loss has anyways stabilized. This can cause a high relative saturation threshold to be too strict in the later stages of training, while it would perform well early on. To handle this, we used a simple strategy where we choose a relative saturation threshold initially, and first time the saturation threshold is crossed, we switch to an absolute threshold with the current loss drop date as the absolute saturation threshold. This essentially amounts to using the relative threshold to determine a good absolute threshold value for the current model and dataset, which is then used subsequently.

Note that, although both *explore phase* and *saturation threshold* prefer higher learning rates, they serve different purposes. While the explore phase ensures that the optimization reaches neighborhood of a wide minima before it reduces the learning rate (by forcing the optimizer to use a higher learning rate for some minimum time, possibly even after it has saturated); saturation threshold adds an overall preference for a higher learning rate until that learning rate has served its purpose and is not optimizing the loss satisfactorily.

## 2.2 HANDLING STOCHASTICITY

For computation of the quadratic coefficients in equation 2, we need to compute the loss values $\hat{L}(\epsilon)$ at multiple values of $\epsilon$. Note that in a typical DNN setting we never compute the full loss, but only a stochastic loss based on a given minibatch. This loss, however, can be noisy and throw off our estimate of $\epsilon_{min}$. To handle this we follow two simple strategies. The first is to use a bigger minibatch (called *superbatch*) for computing the loss values, and the second is to use the *same* superbatch for computing all the losses for a particular estimate. The same strategy is used when computing loss drop rates for the saturation threshold check. We use an integer multiple of minibatches to make a superbatch. This allows us to compute the superbatch loss by simply averaging multiple single batch forward passes, and does not increase the peak GPU memory usage. We use a conservative superbatch size of 100 for all our examples. See section B for details on selection of superbatch size.

Finally, as a safeguard we also added a *rollback* policy. In case our system makes a bad call on the learning rate change (most likely because of a bad superbatch sample), which leads to a reduction in loss drop rate, we rollback the decision and revert the state of the network to the time we made the learning rate change. Although, *rollback* triggers rarely, it is helpful in preventing the optimization from going astray because of one bad decision.

## 3 EXPERIMENTS

We extensively evaluate our method on multiple networks and datasets, as well as multiple optimizers including SGD, Momentum and Adam. We have implemented *AutoLR* as an optimizer in PyTorch (Paszke et al. (2017)), which wraps an existing optimizer, such as SGD, Adam, etc. For our experiments, we used an out of the box policy as in Rolinek & Martius (2018), where we only change the optimizer to *AutoLR*, and don't modify anything else. We evaluate on multiple image

datasets – Imagenet on Resnet-50, Cifar-10 on Resnet-18, MNIST, FashionMNIST; as well as NLP datasets – Squad v1.1 for BERT finetuning and IWSLT with transformer networks. (Note: The source code for *AutoLR* will be open sourced, and a link to it has been sent to Reviewers and ACs for this submission.)

### 3.1 IMAGENET IMAGE CLASSIFICATION ON RESNET-50

In this experiment we trained the ImageNet dataset (Russakovsky et al. (2015)) on Resnet-50 network [3]. We evaluated our method on SGD with momentum which performs best for this dataset. For baseline runs, we used the standard hand-tuned learning rate schedule of $10^{-1}, 10^{-2}, 10^{-3}$ for 30 epochs each. With *AutoLR*, we trained the network with 25 explore epochs, and used the same seed learning rate as baseline, i.e. 0.1. Table 3 shows the training loss and test accuracies for the various runs. As shown, we comfortably beat the test accuracy of the baseline. Figure 4a shows the learning rate and test top-5 accuracy of *AutoLR* against that of the baseline. See figure 9, for more detailed comparisons of training loss, test accuracy, and learning rate.

| LR Schedule | Training Loss | Test Top 1 Acc. | Test Top 5 Acc. |
| --- | --- | --- | --- |
| Baseline | 0.74 (0.001) | 75.87 (0.035) | 92.90 (0.015) |
| *AutoLR* | 0.74 (0.041) | 76.04 (0.098) | 93.01 (0.024) |

Table 3: Training loss and Test accuracy for ImageNet on Resnet-50. We report the mean and standard deviation over 3 runs.

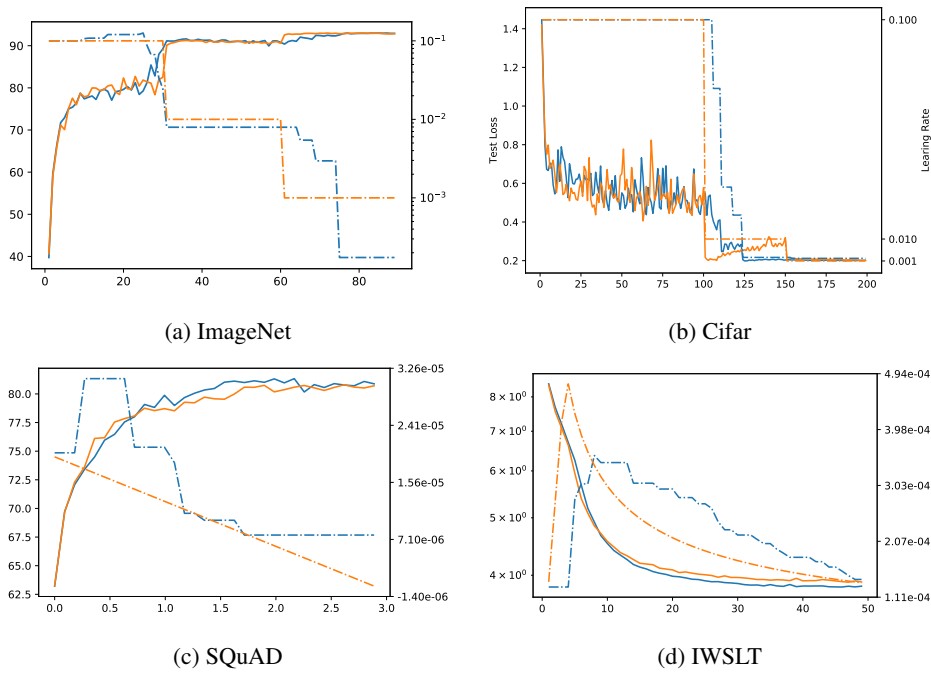

(a) ImageNet

(b) Cifar

(c) SQuAD

(d) IWSLT

Figure 4: Generalization accuracies and learning rates. *AutoLR* plots are show in blue, and baseline in orange. Learning rates are plotted with dashed lines with y-scale on the right, while generalization accuracies are plotted with solid lines with y-scale on the left. The generalization metrics plotted are (a) test top-5 accuracy (b) test accuracy (c) test EM score, and (d) validation perplexity.

### 3.2 CIFAR-10 IMAGE CLASSIFICATION ON RESNET-18

In this experiment we trained the Cifar-10 dataset (Krizhevsky et al. (2009)) on Resnet-18 network (He et al. (2016)) [4]. We evaluated our method on SGD, SGD with momentum and Adam optimizers. For baseline runs of SGD and Momentum, we used a hand-tuned learning rate schedule of

---

[3]We used the implementation at: https://github.com/cybertronai/imagenet18_old
[4]We used the implementation at: https://github.com/kuangliu/pytorch-cifar

$10^{-1}, 10^{-2}, 10^{-3}$ for 100, 50, 50 epochs, respectively; while for Adam we used a fixed learning rate of 1e-3. With *AutoLR*, we trained the network with 100 explore epochs, and used the same seed learning rate as baseline, i.e. 0.1 for SGD, Momentum, and $10^{-3}$ for Adam. Table 4 shows the training loss and test accuracy for the various runs. We also compare against other learning rate schedules such as one-cycle, linear decay and cosine decay for the momentum optimizer in Table 9. Figure 4b shows the learning rate and test loss of *AutoLR* against that of the baseline for Momentum optimizer. See figures 10, 11, 12 for more detailed comparisons of training loss, test accuracy, and learning rate for SGD, Momentum and Adam optimizers, respectively.

| LR Schedule | Optimizer | | |
|---|---|---|---|
| | SGD | Momentum | Adam |
| Training Loss | | | |
| Baseline | 4.6e-4 (3.3e-5) | 0.002 (6.5e-5) | 0.005 (8e-4) |
| *AutoLR* | 7e-4 (1.1e-4) | 8e-4 (1e-4) | 0.004 (4e-4) |
| Test Accuracy | | | |
| Baseline | 92.63 (0.002) | 94.81 (0.001) | 92.46 (0.002) |
| *AutoLR* | 92.70 (0.004) | 94.79 (0.001) | 92.75 (0.005) |

Table 4: Training loss and Test accuracy for Cifar-10 on Resnet-18. We report the mean and standard deviation over 7 runs.

See sections G.1 and section G.2 for results on MNIST and FashionMNIST datasets with SGD, Momentum and Adam optimizers.

### 3.3 SQuAD FINE-TUNING ON BERT

We now evaluate *AutoLR* on a few NLP tasks. In the first task, we fine-tune the $\text{BERT}_{\text{BASE}}$ model (Devlin et al. (2018)) on SQuAD v1.1 (Rajpurkar et al. (2016)) [5]. BERT fine-tuning is prone to ovefitting because of a huge model size compared to the fine-tuning dataset, and is typically run for only a few epochs. We use the standard baseline which trains for 3 epochs with a seed learning rate of $2e - 5$ with linear decay. With *AutoLR* we found that the model overfits much earlier, seemingly because *AutoLR* reduces the training loss much more quickly (see figure 14), and we thus needed to fine-tune for only 2 epochs. The *AutoLR* runs were trained with 2500 explore steps ($\approx$ half epoch), and the same seed learning rate of $2e - 5$ as baseline. Table 5 shows our results over 3 runs. We achieve a best EM score of 81.2 (after 2 epochs), compared to baseline's best of 80.7 (after 3 epochs). We found it interesting that *AutoLR* was able to tune the learning rate effectively even in the small budget of 2 epochs to beat the baseline number of $\text{BERT}_{\text{BASE}}$ (see Table-2 of Devlin et al. (2018) who reported an EM score of 80.8). Figure 4c shows the learning rate and test EM score of *AutoLR* against that of the baseline. See figure 14 for detailed comparisons. We also compare against other learning rate schedules such as one-cycle, linear decay and cosine decay in Table 11.

| LR Schedule | Train Loss (av) | EM (best) | EM (av) | F1 (av) |
|---|---|---|---|---|
| Baseline | 0.96 (0.075) | 80.7 | 80.4 (0.18) | 88.2 (0.02) |
| *AutoLR* | 1.05 (0.008) | 81.2 | 80.8 (0.52) | 88.5 (0.09) |

Table 5: SQuAD fine-tuning on BERT. We report the average training loss, best test EM score, and average test EM, F1 scores over 3 runs.

### 3.4 MACHINE TRANSLATION ON TRANSFORMER NETWORK WITH IWSLT

In the second NLP task, we train the Transformer network (Vaswani et al. (2017)) on the IWSLT German-to-English (De-En) dataset (Cettolo et al. (2014)) with the Adam optimizer [6]. For baseline, we used the learning rate schedule mentioned in Vaswani et al. (2017). The baseline learning rate starts at $1.25e - 7$, and is linearly increased for 4000 steps, followed by an inverse square root decay till 50 epochs. With *AutoLR*, we trained the network with 8 explore epochs, and used a seed learning

---

[5]We used the implementation at: https://github.com/huggingface/pytorch-transformers
[6]We used the implementation at: https://github.com/pytorch/fairseq

rate of 5e-5. In both cases we use the model checkpoint with least loss on the validation set for computing BLEU scores on the test set. Table 6 shows the training loss and test accuracy averaged over 3 runs. As shown, *AutoLR* achieves a mean test BLEU score of 34.88, compared to 34.70 for the baseline. Figure 4d shows the learning rate and validation perplexity of *AutoLR* against that of the baseline. See figure 13 for detailed comparisons of training/validation perplexity, learning rate, etc. We also compare against other learning rate schedules such as one-cycle, linear decay and cosine decay in Table 10. It is interesting to note that both linear decay and cosine decay schedules outperform the baseline inverse square root decay schedule of Vaswani et al. (2017).

| LR Schedule | Train ppl | Validation ppl | Test BLEU Score |
|---|---|---|---|
| Baseline | 3.55 (0.029) | 5.10 (0.033) | 34.70 (0.001) |
| *AutoLR* | 3.46 (0.16) | 4.86 (0.014) | 34.88 (0.005) |

Table 6: Training, validation perplexity and test BLEU scores for IWSLT on Transformer networks. The test BLEU scores are computed on the checkpoint with the best validation perplexity. We report the mean and standard deviation over 3 runs.

Section E has more details of the one-cycle, cosine decay and linear decay baselines. Table 7 shows the test accuracy of different experiments on all learning rate schedules tried in this paper. As shown, *AutoLR* compares favorably against all other learning rate schedules, except slightly missing out in Cifar-10. For one-cycle we report results from the default policy of finding the maximum learning rate, as well as by hand tuning it (see section E for more details). The Trapezoid schedule follows the same seed learning rate and max learning rate as the tuned values of the corresponding one-cycle run. Similarly, the default cosine decay and linear decay schedules did not work well for IWSLT, so we added a warmup phase in the beginning as is done in the original paper's baseline. We report both these results under the default and tuned columns, respectively. An interesting observation is that both cosine and linear decay perform better than the inverse square root decay learning rate schedule suggested in the "Attention is all you need" paper. A similar observation can be made for the Bert fine-tuning for SQuAD. The fact that the original authors used a suboptimal learning rate points to the general difficulties in learning rate tuning. More importantly, these results show that for fixed function classes of learning rate schedules, it is not just important to tune the hyperparameters of each function class, but also try out different function classes. Since the space of function classes is huge in itself, searching for the best schedule is typically intractable.

| Experiment | *AutoLR* | Baseline | Trapezoid | One-Cycle | | Cosine Decay | | Linear Decay | |
|---|---|---|---|---|---|---|---|---|---|
| | | | | default | tuned | default | tuned | default | tuned |
| ImageNet | **93.01** | 92.90 | - | - | - | - | - | - | - |
| Cifar-10 | 94.79 | **94.81** | 94.69 | 93.38 | 94.59 | 94.52 | - | 94.24 | - |
| IWSLT | **34.88** | 34.70 | 34.85 | 32.35 | 34.57 | 0.33 | 34.85 | 0.28 | 34.82 |
| SQuAD | **81.2** | 80.7 | 80.1 | 79.8 | 80.9 | 80.8 | - | 80.7 | - |

Table 7: Test accuracy numbers for different experiments on all learning rate schedules tried in this paper. We report the average top-5 accuracy for ImageNet, average top-1 accuracy for Cifar-10, average BLEU score for IWSLT and the best EM score for SQuAD. The ImageNet results for one-cycle, cosine-decay, linear-decay are still running and will be updated in an updated draft.

## 4 RELATED WORK

**Generalization.** There has been a lot of work recently on understanding the generalization characteristics of DNNs. Kawaguchi (2016) found that DNNs have many local minima, but all local minima were also the global minima. It has been observed by several authors that wide minima generalize better than narrow minima (Arora et al. (2018); Hochreiter & Schmidhuber (1997); Keskar et al. (2016); Jastrzebski et al. (2017); Wang et al. (2018)) but there have been other work questioning this hypothesis as well (Dinh et al. (2017); Golatkar et al. (2019); Guiroy et al. (2019); Jastrzebski et al. (2019); Yoshida & Miyato (2017)).

Keskar et al. (2016) found that small batch SGD generalizes better and lands in wider minima than large batch SGD. However, recent work has been able to generalize quite well even with very large

batch sizes (Goyal et al. (2017); McCandlish et al. (2018); Shallue et al. (2018)), by scaling the learning rate linearly as a function of the batch size. Jastrzebski et al. (2019) analyze how batch size and learning rate influence the curvature of not only the SGD endpoint but also the whole trajectory. They found that small batch or large step SGD have similar characteristics, and yield smaller and earlier peak of spectral norm as well as smaller largest eigenvalue. Dinh et al. (2017) show analytically using model reparameterization that wide minima can be converted to sharp minima without hurting generalization. Wang et al. (2018) analytically show that generalization of a model is related to the Hessian and propose a new metric for the generalization capability of a model that is unaffected by model reparameterization of Dinh et al. (2017). Yoshida & Miyato (2017) argue that regularizing the spectral norm of the weights of the neural network help them generalize better. On the other hand, Arora et al. (2018) derive generalization bounds by showing that networks with low stable rank (high spectral norm) generalize better. Guiroy et al. (2019) looks at generalization in gradient-based meta-learning and they show experimentally that generalization and wide minima are not always correlated. Finally, Golatkar et al. (2019) show that regularization results in higher test accuracy specifically when it is applied during initial phase of training, similar to the importance of *AutoLR*ś explore phase during initial phase of training.

**Lower density of wide minima.** Wu et al. (2018) compares the sharpness of minima obtained by batch gradient descent (GD) with different learning rates for small neural networks on FashionM-NIST and Cifar10 datasets. They find that GD with a given learning rate finds the theoretically sharpest feasible minima for that learning rate. Thus, in the presence of several flatter minimas, GD with lower learning rates does not find them, leading the authors to conjecture that density of sharper minima is perhaps larger than density of wider minima.

**Adaptive learning.** Many adaptive learning rate methods have been developed which adapt the learning rate on a per parameter basis, including AdaGrad (Duchi et al. (2011)), ADADELTA (Zeiler (2012)), RMSProp (Tieleman & Hinton (2012)), and Adam (Kingma & Ba (2014)). However, most of these methods still require specifying a global learning rate schedule which impacts the performance significantly. Many methods have been proposed to adaptively tune the global learning rate, such as Schaul et al. (2013a) which makes an idealized quadratic loss function assumption near the minima, Baydin et al. (2017) which uses gradient w.r.t the learning rate to update the learning rate in a first order fashion, and Rolinek & Martius (2018) which again uses a first order method along with an estimate of minimum loss achievable in a step. While Schaul et al. (2013a) and Baydin et al. (2017) compare to only constant learning rate baselines, Rolinek & Martius (2018) does compare to state-of-the-art learning rate schedule for Cifar-10, but is unable to match their test accuracy, although does better on train loss. Finally, approaches like one cycle learning (Smith (2018); Xing et al. (2018)) have been proposed that linearly increase and decrease the learning rate in a specific manner. Based on our evaluation, one cycle scheme works reasonably well with manual tuning of the learning rate but does not always achieve state-of-the-art accuracy.

## 5 CONCLUSIONS

We have presented *AutoLR*, an *Explore-Exploit* method for auto learning rate tuning for training deep neural networks. Our *explore* phase is based on the hypothesis that narrow minimas far outnumber the wide minimas, and thus require some minimum exploration at a high learning rate to land in a wide minima region with high probability. We do multiple validations of this hypothesis, but also plan to further study it both theoretically and empirically in future work. Our *exploit* phase is based on a local quadratic approximation of loss in the search direction, as a function of perturbations in the learning rate. We extensively validate *AutoLR* on both image (ImageNet, Cifar-10, MNIST, FashionMNIST) and NLP (IWSLT, Squad) datasets on as well as multiple optimizers, and achieve or beat generalization accuracy of state of the art hand tuned learning rate schedules.

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

## A    VALIDATION OF QUADRATIC APPROXIMATION

We want to validate whether our quadratic approximation is a good approximation for modelling the loss as a function of the perturbation $\epsilon$. Figure 5 shows a few examples demonstrating the effectiveness of second order approximation. As shown, the loss values at various samples overlap with the quadratic curve almost perfectly, including those not used in estimation of the quadratic (orange triangles). Also, the estimated loss value at the quadratic minima matches the true loss value there quite well. Figure 6 shows quadratic plots for more datasets/models.

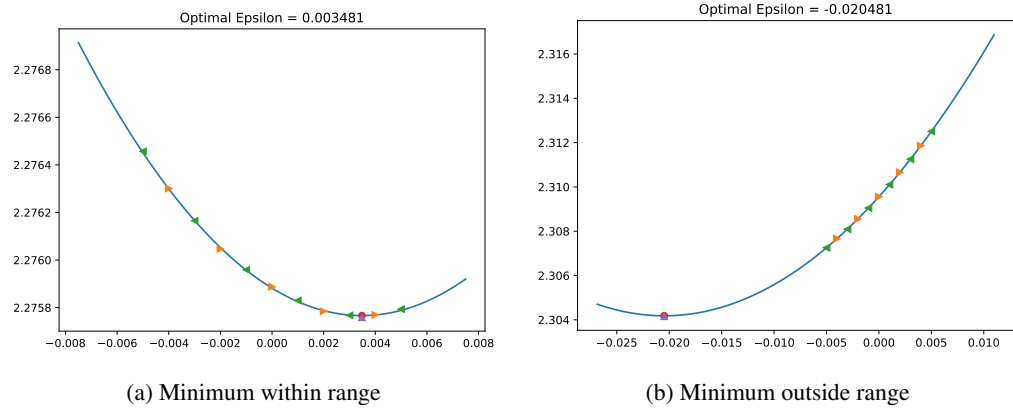

(a) Minimum within range                    (b) Minimum outside range

Figure 5: Quadratic approximation of loss as a function of $\epsilon$. Shown are two examples from Cifar-10 on Resnet-18 runs where the minimum is (a) within and (b) outside the range of loss samples. The orange triangles show loss samples used in fitting the quadratic, the blue line shows our quadratic approximation, and the green triangles show more loss samples which were not used for fitting. The red circle shows the minimum loss value as per the quadratic, while the purple triangle show the true value at that $\epsilon$.

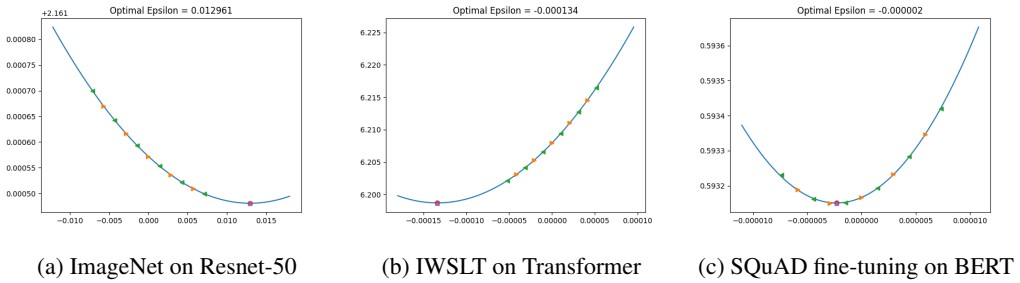

(a) ImageNet on Resnet-50          (b) IWSLT on Transformer          (c) SQuAD fine-tuning on BERT

Figure 6: Quadratic approximation curve samples from (a) ImageNet on Resnet-50, (b) IWSLT on Transformer and (c) SQuAD fine-tuning on BERT. The legend is same as figure 5. It can be seen that the quadratic approximation works pretty well.

## B    SUPERBATCH SIZE SELECTION

To choose an appropriate superbatch size, we measure the standard deviation of loss as a function of superbatch size. We compute this by evaluating the loss with 10 different randomly sampled superbatches and calculate the standard deviation. Figure 7 shows the measurements. We used a conservative superbatch size of 100 in all our examples, as it corresponded to low variance. Note that higher superbatch sizes add an increased computational overhead, but since we recompute our learning rate infrequently the total overhead is not very high.

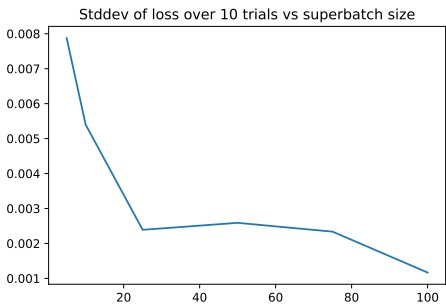 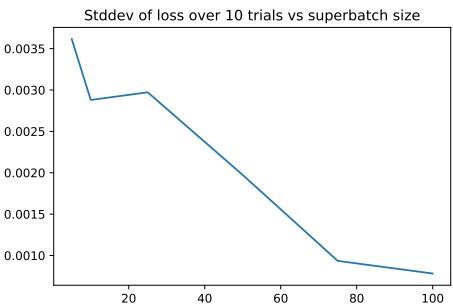

Figure 7: Standard deviation of loss as a function of superbatch size for (a) Cifar10 on Resnet-18, and (b) MNIST.

## C    COMPUTATIONAL COST

The primary computational cost of our method comes from computing the samples for quadratic approximation. We use a superbatch size of 100 and pick 5 samples for approximating our quadratic, thus incurring a cost of 500 forward passes each time we need to recompute the learning rate. Note that we recompute the learning rate only once in *recompute_window* steps. In most of our examples we keep the *recompute_window* such that our learning rate recomputation occurs 2 times each epoch, except in BERT fine-tuning where we only have 2-3 epochs to train, and thus want more frequent tuning of the learning rate. In Cifar-10 on Resnet-18 for example, we used a *recompute_window* of 1000 minibatch steps. A backward pass is typically 2x more computationally expensive than forward pass, and 1000 minibatch steps thus cost 3000 forward pass worth of compute. Our recomputation costs 500 forward passes of compute, and thus the computational overhead is $500/3000 \approx 16.6\%$. Note, however that in *exploit* phase, we do a recomputation of learning rate only when saturation threshold is crossed. So the computational cost comes down even further and $16.6\%$ overhead for Cifar-10 is an upper bound.

Note that since the main focus of this work was to develop an automatic learning rate tuning scheme which generalizes as well as state of the art hand tuned learning rate schedules, we have not invested much effort on reducing the computational cost. For example, a 100 superbatch size is an overkill and a size of 50 should be suitable for most examples. Similarly, 3 samples for estimating the quadratic are enough most of the times instead of the 5 used, as the quadratic fit is very close mostly. Also, we can experiment with higher *recompute_window* sizes to reduce the cost further.

## D    HYPERPARAMETERS

Table 8 shows the hyperparameters used for all our examples. As shown, the saturation threshold and epsilon thresholds are pretty stable across all image examples, and the main parameter to tune is the explore epochs, which we typically set to around $20 - 30\%$ of the total budget.

The reason for different scale of epsilon thresholds in NLP datasets compared to image datasets, is because they typically run on much lower learning rates (of the order of $1e - 5$) compared to image datasets (of the order of $1e - 1$ to $1e - 3$), which suggests that the optimization landscapes are more sensitive to smaller perturbations and thus need more aggressive clipping. We have some initial ideas on computing this threshold dynamically by looking at change in loss magnitude as we increase the $\epsilon$, and choosing a threshold which limits this change magnitude to some amount. This can be computationally more expensive, as we need to search for the right epsilon, but can be done only few times in the each run.

We also noticed that the loss drop rate changes much more drastically in image datasets than NLP datasets, again most likely because of the very low learning rates used in NLP datasets. Thus we need to use a lower saturation threshold in NLP experiments compared to image ones. We found this hyperparameter very easy to set, by simply eyeballing at the loss drop rate changes of a trial run with fixed LR.

We are also looking at ways to automate the explore epochs hyperparameter, by looking at second order information about the loss landscape, etc. to determine if we have reached a wider minima region. See Jastrzebski et al. (2019) for some interesting analysis on this front.

| Experiments | Explore Epochs / Total Epochs | Saturation Threshold | Epsilon Threshold |
|---|---|---|---|
| MNIST | 10 / 50 | 100 | 1e-3 |
| Fashion-MNIST | 10 / 50 | 100 | 1e-3 |
| Cifar-10 | 100 / 200 | 100 | 5e-3 |
| Imagenet | 25 / 90 | 500 | 1e-3 |
| IWSLT'14 (De-En) | 8 / 50 | 5 | 1e-9 |
| SQuADv1.1 (Bert-Base) | 0.45 / 2 | 2 | 1e-12 |

Table 8: Hyperparameters used for all experiments.

## E    COMPARISONS WITH MORE BASELINE LEARNING RATE SCHEDULES

In this section we compare *AutoLR* against more learning rate schedules – one-cycle, linear decay and cosine decay.

**One-Cycle**: The one-cycle learning rate schedule was proposed in Smith (2018) (also see Smith (2017)). This schedule first chooses a maximum learning rate based on an LR Range test. The LR range test starts from a small learning rate and keeps increasing the learning rate until the loss starts exploding. Smith (2018) suggests that the maximum learning rate should be chosen to be bit before the minima, in a region where the loss is still decreasing. There is subjectivity in making this choice, although some blogs and libraries[7] suggest using a learning rate one order lower than the one at minima. However, in our experience, we found that this choice did not perform well. Instead, we found that values closer to the minima performed much better. In the experiments below we tried multiple values closer to the minima where the loss was still decreasing and report the best results as *one-cycle (tuned)*. Results with the default policy of $1/10^{th}$ the minima are reported as *one-cycle (default)*.

Once the maximum learning rate is chosen, the one-cycle schedule proceeds as follows. The learning rate starts at a specified fraction[8] of the maximum learning rate and is increased linearly to the maximum learning rate for 45 percent of the training budget and then decreased linearly for the remaining 45. For the final 10 percent, the learning rate is reduced by a large factor (we chose a factor of 10). For the Momentum optimizer, Smith (2018) suggests to reduce the momentum linearly when lr is increasing, and vice versa. We used an opensource implementation [9] for our experiments.

**Trapezoid**: The trapezoid learning rate schedule was proposed in Xing et al. (2018), and is a variation on the one-cycle schedule. The trapezoid schedule increases the learning rate as in the first phase of one-cycle, but then keeps it flat at that value for sometime, before linearly decaying it again like the second phase of one-cycle. Following the Trapezoid (long) schedule of Xing et al. (2018) (see Figure 7,8 there), in all our experiments we linearly warmup for the first $10\%$ of iterations, stay flat for the next $80\%$, and then linearly decay for the final $10\%$ iterations. For the maximum learning rate we use the values of the tuned one-cycle runs.

**Linear Decay**: The linear decay learning rate schedule simply decays the learning rate linearly to zero starting from a seed LR. In the case of IWSLT we found that starting from a high seed caused instability issues (most likely because of Adam optimizer. See Liu et al. (2019)). In this case we also ran experiments with a linear warmup followed by this linear decay.

---

[7]See e.g. `https://towardsdatascience.com/finding-good-learning-rate-and-the-one-cycle-policy-7` and `https://sgugger.github.io/how-do-you-find-a-good-learning-rate.html`. Also see `https://docs.fast.ai/callbacks.lr_finder.html` and `https://docs.fast.ai/callbacks.one_cycle.html`

[8]See div_factor in `https://docs.fast.ai/callbacks.one_cycle.html`. We chose the fraction to be 0.1 in our experiments.

[9]`https://github.com/nachiket273/One_Cycle_Policy`

**Cosine Decay**: The cosine decay learning rate schedule decays the learning rate to zero following a cosine curve, starting from a seed LR. Similar to Linear decay, in the case of IWSLT, we found that starting from a high seed caused instability issues. In this case we also ran experiments with a linear warmup followed by this cosine decay.

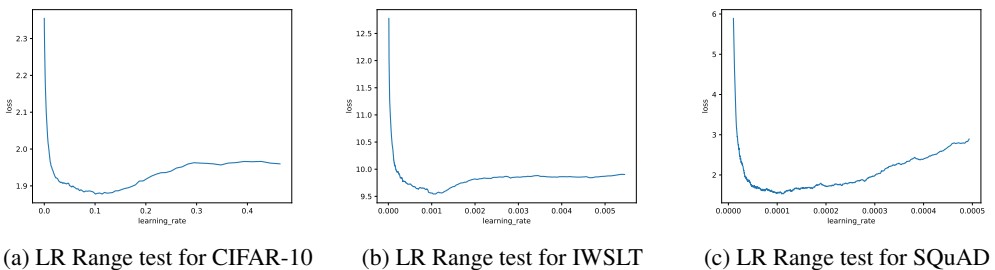

(a) LR Range test for CIFAR-10   (b) LR Range test for IWSLT   (c) LR Range test for SQuAD

Figure 8: LR Range test for selecting the maximum learning rate. A good choice is the learning rate is a bit before the minima in a region where the loss is still decreasing.

### E.1    CIFAR-10

Figure 8a shows the LR range test for Cifar-10 with the Resnet-18 network. The minima occurs around learning rate of 0.09. For the maximum learning rate, we choose 9e-3 for the default one-cycle policy, but found 0.06 to perform best in our tuning. For linear, cosine decay schedules we start with a seed learning rate of 0.1 as used in the standard baselines. Table 9 shows the training loss and test accuracy for the various schedules.

| LR Schedule | Train Loss | Test Accuracy |
|---|---|---|
| Trapezoid | 0.0009 (8.7e-5) | 94.69 (0.0007) |
| One-Cycle (default) | 0.0003 (3e-5) | 93.38 (0.001) |
| One-Cycle (tuned) | 0.0004 (3.1e-5) | 94.59 (0.0007) |
| Cosine Decay | 0.0003 (2.7e-5) | 94.52 (0.0016) |
| Linear Decay | 0.0002 (2.5e-5) | 94.24 (0.0009) |
| *AutoLR* | 0.0008 (1.0e-4) | 94.79 (0.001) |

Table 9: Training loss and Test accuracy for Cifar-10 on Resnet-18 for more learning rate schedules. We report the mean and standard deviation over 7 runs. For One-Cycle, we report experiments with maximum learning rate chosen via the default lr-range finder policy and via manual tuning. The Trapezoid schedule uses maximum learning rate of the tuned setting.

### E.2    IWSLT'14 DE-EN

Figure 8b shows the LR range test for IWSLT on the transformer networks. The minima occurs at 9e-4. For the maximum learning rate, we choose 9e-5 for the default one-cycle policy and found 2e-4 to perform best in our tuning. For linear, cosine decay schedules we first used a seed learning rate of 5e-4 (the maximum learning rate in the schedule in Vaswani et al. (2017)). However, we found this to perform very poorly, most likely due to initial instabilities in Adam as discussed in Liu et al. (2019). So we also ran experiments with linear warmup as in Vaswani et al. (2017) followed by linear/cosine decays which performed much better. Table 10 shows the training, validation perplexity and BLEU scores for the various schedules.

### E.3    SQUAD-v1.1

Figure 8c show the LR range test for SQuAD fine-tuning on BERT. The minima occurs at 1e-4. For the maximum learning rate, we choose 1e-5 for the default one-cycle policy and found 4e-5 to perform best in our tuning. For linear, cosine decays we start with a seed learning rate of 2e-5 as used in Devlin et al. (2018). Table 11 show the average training loss, best test EM score, and average test EM, F1 scores for the various schedules.

| LR Schedule | Train ppl | Validation ppl | Test BLEU Score |
|---|---|---|---|
| Trapezoid | 3.25 (0.016) | 4.92 (0.014) | 34.85 (0.008) |
| One-Cycle (default) | 5.51 (0.028) | 5.87 (0.02) | 32.35 (0.18) |
| One-Cycle (tuned) | 3.82 (0.09) | 5.01 (0.03) | 34.57 (0.002) |
| Cosine Decay | 9.54 (0.06) | 3861.6 (883.4) | 0.33 (1e-5) |
| Cosine Decay with warmup | 3.29 (0.02) | 5.04 (0.019) | 34.85 (0.008) |
| Linear Decay | 9.50 (0.07) | 3261.5 (849.7) | 0.28 (2e-5) |
| Linear Decay with warmup | 3.30(0.02) | 5.03 (0.016) | 34.82 (0.006) |
| *AutoLR* | 3.46 (0.16) | 4.86 (0.014) | 34.88 (0.005) |

Table 10: Training, validation perplexity and test BLEU scores for IWSLT on Transformer networks for more learning rate schedules. The test BLEU scores are computed on the checkpoint with the best validation perplexity. We report the mean and standard deviation over 3 runs.

| LR Schedule | Train Loss (av) | EM (best) | EM (av) | F1 (av) |
|---|---|---|---|---|
| Trapezoid | 0.95(0.002) | 80.2 | 80.1 (0.05) | 87.9 (0.04) |
| One Cycle (default) | 1.364 (0.011) | 79.8 | 79.4 (0.17) | 87.3 (0.12) |
| One Cycle (tuned) | 1.016 (0.004) | 80.9 | 80.7 (0.12) | 88.4 (0.09) |
| Cosine Decay | 0.89 (0.003) | 80.8 | 80.6 (0.06) | 88.4 (0.13) |
| Linear decay | 0.96 (0.075) | 80.7 | 80.4 (0.18) | 88.2 (0.02) |
| *AutoLR* | 1.05 (0.008) | 81.2 | 80.8 (0.52) | 88.5 (0.09) |

Table 11: SQuAD fine-tuning on BERT for more learning rate schedules. We report the average training loss, best test EM score, and average test EM, F1 scores over 3 runs.

## F  SEED SENSITIVITY

We performed a sensitivity analysis on the importance of the choice of seed learning rate for *AutoLR*. We ran *AutoLR* for Cifar-10 on Resnet-18 with different seeds in a 10x range. Table 12 shows the average test accuracies over 3 runs for the different seeds. As shown, the seed learning rate can impact the final accuracy, but *AutoLR* is not highly sensitive to it, especially in the lower ranges.

| Seed LR | Test Accuracy |
|---|---|
| 0.05 | 94.60 |
| 0.075 | 94.71 |
| 0.1 | 94.79 |
| 0.125 | 94.58 |
| 0.15 | 94.25 |

Table 12: Cifar-10 on Resnet-18 trained for 200 epochs with Momentum. We report averages over 3 runs.

## G  MORE RESULTS

### G.1  MNIST

In this experiment we trained the MNIST dataset LeCun et al. (1998) on a 4 layer network with two convolution layers and two fully connected layers [10]. We evaluated our method on SGD, SGD with momentum and Adam optimizers. For baseline runs, we used a fixed learning rate of 0.1 for SGD, Momentum, and 1e-3 for Adam; and trained for 50 epochs. With *AutoLR*, we trained the network with 10 explore epochs, and used the same seed learning rate as baseline, i.e. 0.1 for SGD, Momentum, and 1e-3 for Adam. Table 13 shows the training loss and test accuracy for the various runs. See figures 15, 16, 17 for more detailed comparisons of training loss, test accuracy, and learning rate for SGD, Momentum and Adam optimizers, respectively.

---

[10]We used the implementation at: https://github.com/pytorch/examples/tree/master/mnist

| LR Schedule | Optimizer | | |
|---|---|---|---|
| | SGD | Momentum | Adam |
| Training Loss | | | |
| Baseline | 2.3e-5 (9.4e-7) | 7.2e-6 (4.2e-7) | 4.5e-3 (1.3e-3) |
| *AutoLR* | 2.4e-5 (5.5e-6) | 7.3e-6 (7.7e-7) | 3.9e-4 (9.5e-4) |
| Test Accuracy | | | |
| Baseline | 99.32 (3.4e-4) | 99.34 (6.0e-4) | 99.21 (1.2e-3) |
| *AutoLR* | 99.32 (3.7e-4) | 99.38 (5.1e-4) | 99.35 (7.2e-4) |

Table 13: Training loss and Test accuracy for MNIST. We report the mean and standard deviation over 7 runs.

### G.2 FASHION MNIST

In this experiment we trained the Fashion MNIST dataset Xiao et al. (2017) on the same 4 layer network as for the MNIST dataset. We evaluated our method on SGD, SGD with momentum and Adam optimizers. For baseline runs, we used a fixed learning rate of 0.1 for SGD, Momentum, and 1e-3 for Adam; and trained for 50 epochs. With *AutoLR*, we trained the network with 10 explore epochs, and used the same seed learning rate as baseline, i.e. 0.1 for SGD, Momentum, and 1e-3 for Adam. Table 14 shows the training loss and test accuracy for the various runs. See figures 18, 19, 20 for more detailed comparisons of training loss, test accuracy, and learning rate for SGD, Momentum and Adam optimizers, respectively.

| LR Schedule | Optimizer | | |
|---|---|---|---|
| | SGD | Momentum | Adam |
| Training Loss | | | |
| Baseline | 0.005 (0.006) | 0.073 (0.007) | 0.018 (0.003) |
| *AutoLR* | 0.001 (0.001) | 4.3e-4 (7.2e-4) | 0.002 (0.006) |
| Test Accuracy | | | |
| Baseline | 91.33 (0.005) | 89.31 (0.003) | 90.56 (0.003) |
| *AutoLR* | 91.88 (0.001) | 91.66 (0.002) | 91.65 (0.003) |

Table 14: Training loss and Test accuracy for FashionMNIST. We report the mean and standard deviation over 7 runs.

# H  DETAILED PLOTS

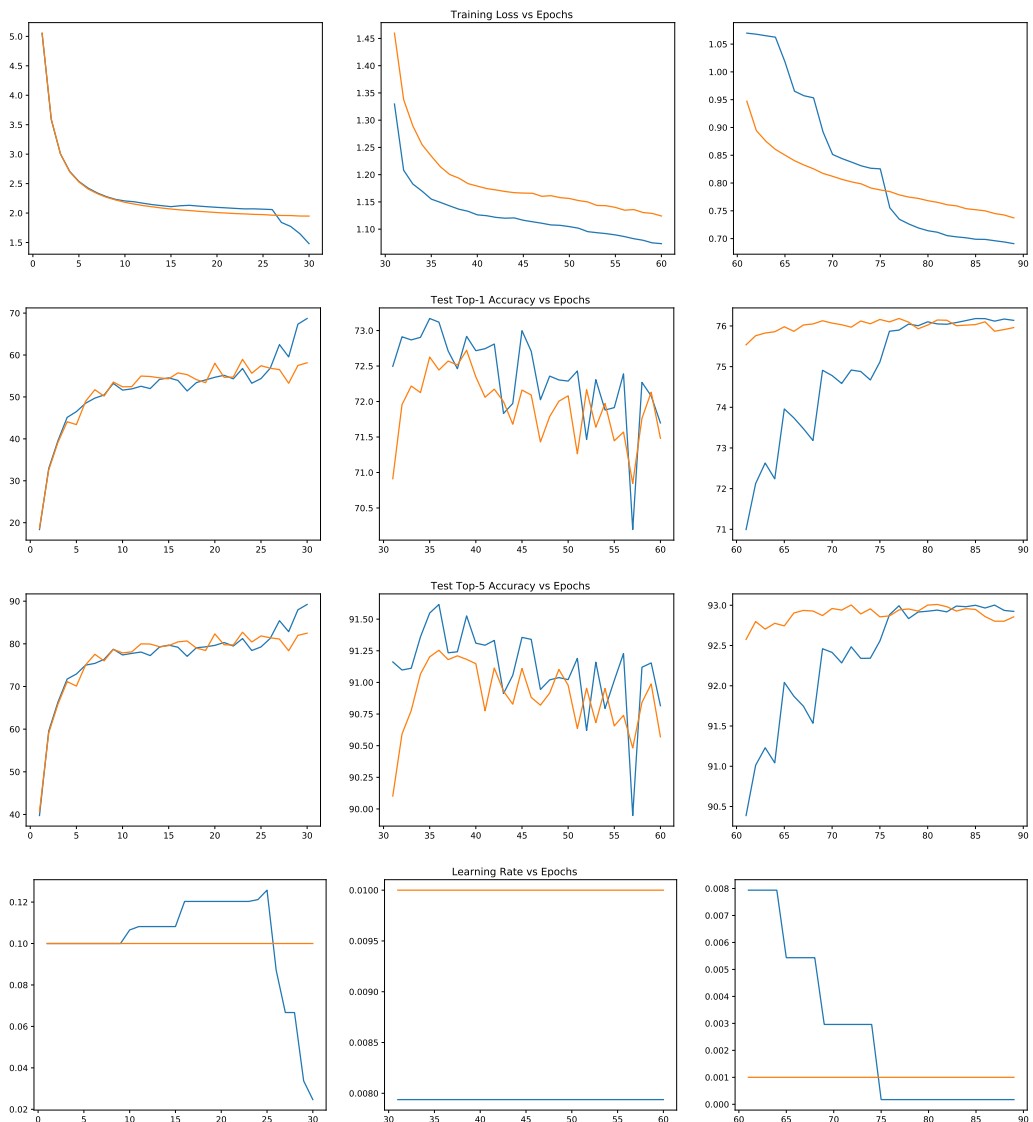

Figure 9: ImageNet on Resnet-50 trained with Momentum. Shown are the training loss, top-1/top-5 test accuracy and learning rate as a function of epochs, for the baseline scheme (orange) vs the *AutoLR* scheme (blue). The plot is split into 3 parts to permit higher fidelity in the y-axis range.

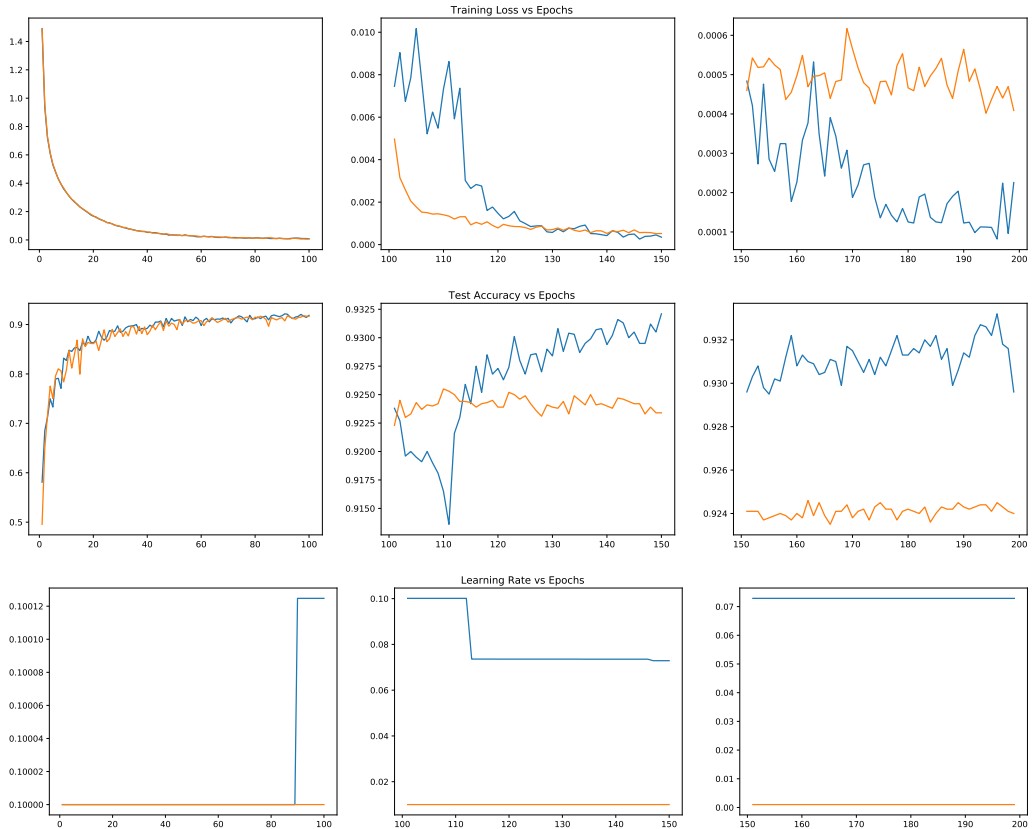

Figure 10: Cifar-10 on Resnet-18 trained with SGD. Shown are the training loss, test accuracy and learning rate as a function of epochs, for the baseline scheme (orange) vs the *AutoLR* scheme (blue). The plot is split into 3 parts to permit higher fidelity in the y-axis range.

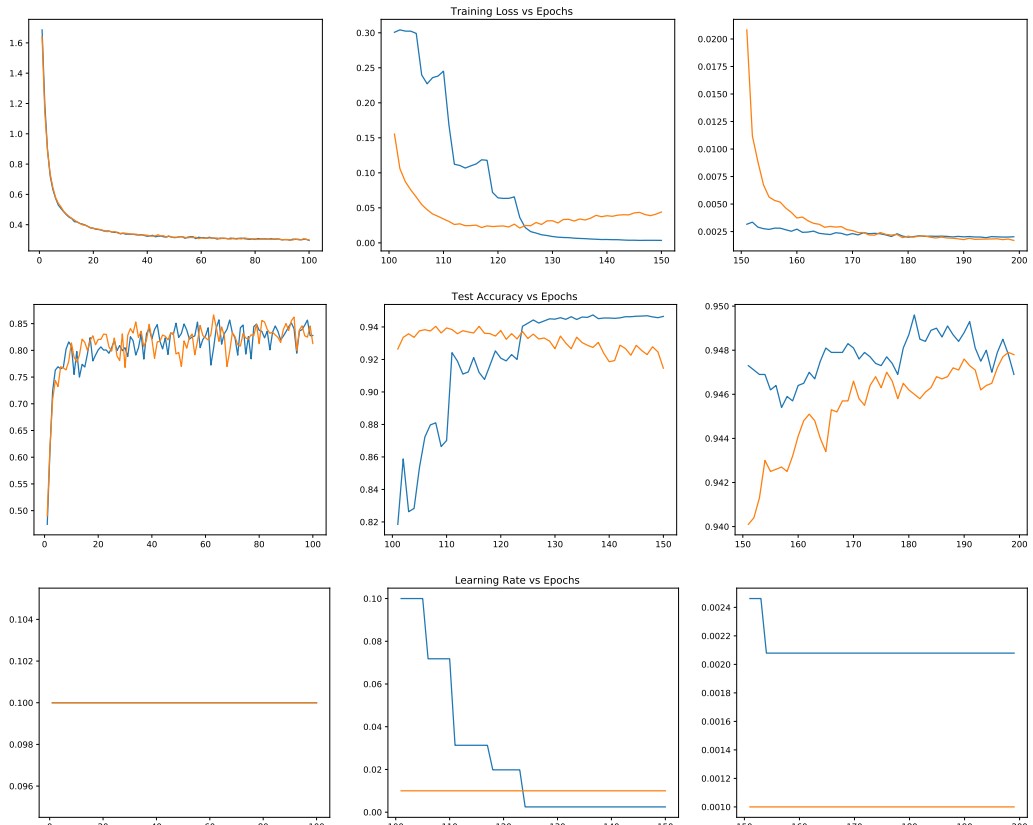

Figure 11: Cifar-10 on Resnet-18 trained with Momentum. Shown are the training loss, test accuracy and learning rate as a function of epochs, for the baseline scheme (orange) vs the *AutoLR* scheme (blue). The plot is split into 3 parts to permit higher fidelity in the y-axis range.

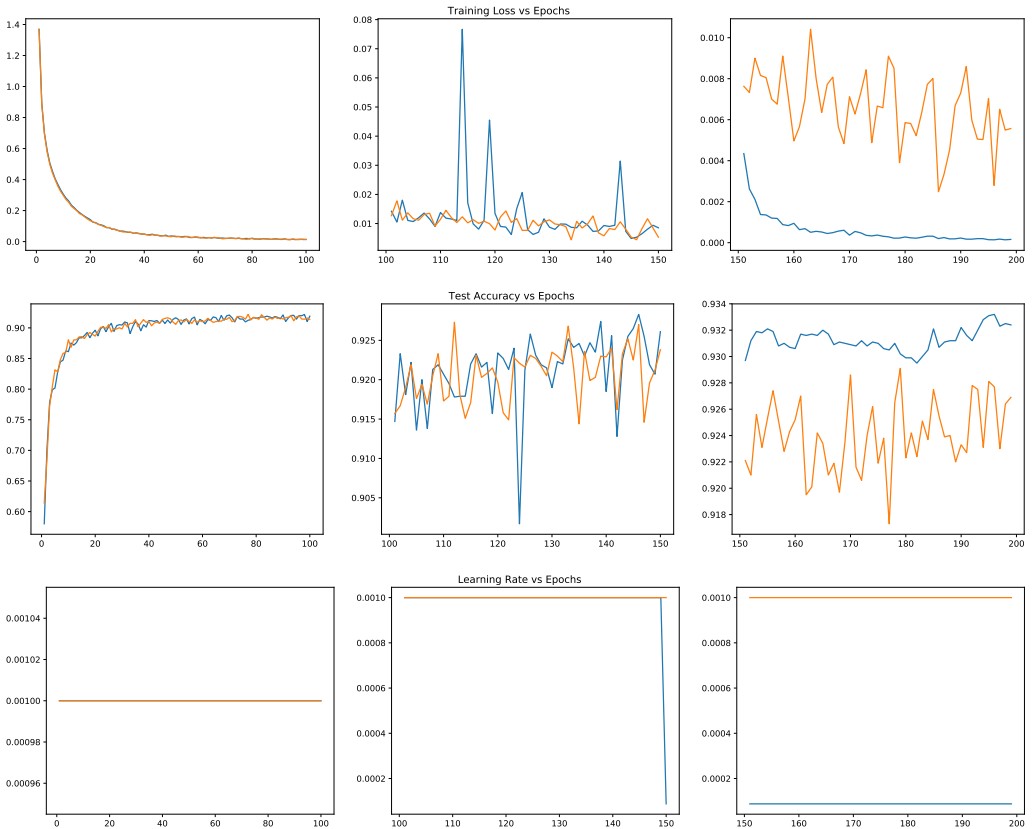

Figure 12: Cifar-10 on Resnet-18 trained with Adam. Shown are the training loss, test accuracy and learning rate as a function of epochs, for the baseline scheme (orange) vs the *AutoLR* scheme (blue). The plot is split into 3 parts to permit higher fidelity in the y-axis range.

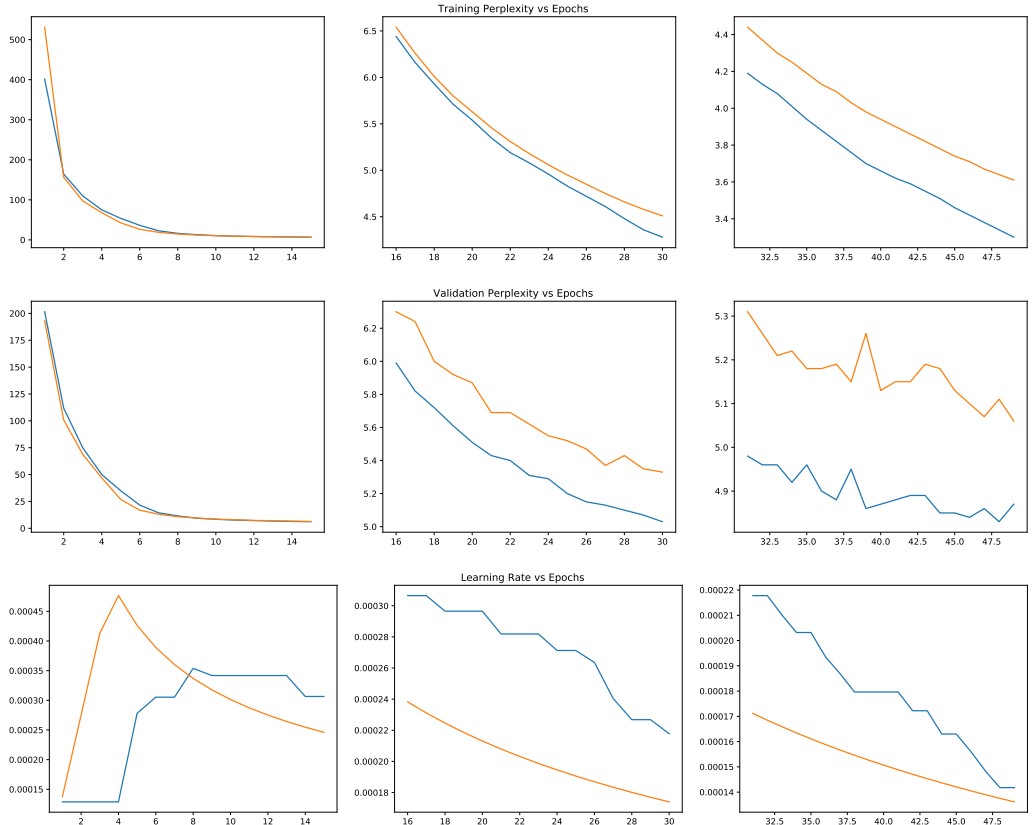

Figure 13: IWSLT on Transformer network trained with Adam. Shown are the training perplexity, validation perplexity and learning rate as a function of epochs, for the baseline scheme (orange) vs the *AutoLR* scheme (blue). The plot is split into 3 parts to permit higher fidelity in the y-axis range.

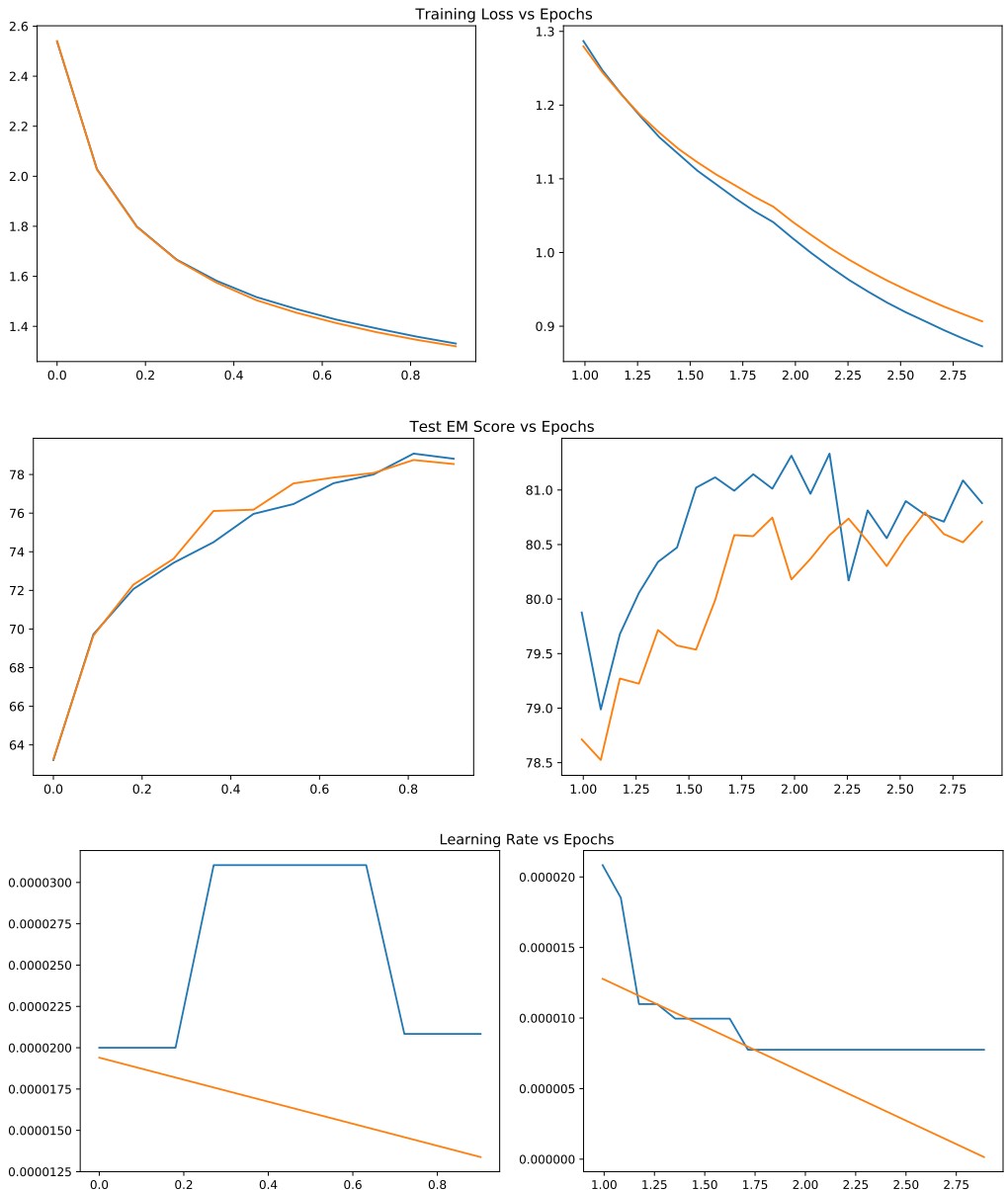

Figure 14: SQuAD fine-tuning on BERT trained with Adam. Shown are the training loss, test EM score, and learning rate as a function of epochs, for the baseline scheme (orange) vs the *AutoLR* scheme (blue). The plot is split into 2 parts to permit higher fidelity in the y-axis range. It is clear that with *AutoLR* the network starts to overfit after the 2nd epoch, where the testing loss continues to go down, but generalization suffers. We saw similar behavior with different seeds, and thus need to train with *AutoLR* for only 2 epochs.

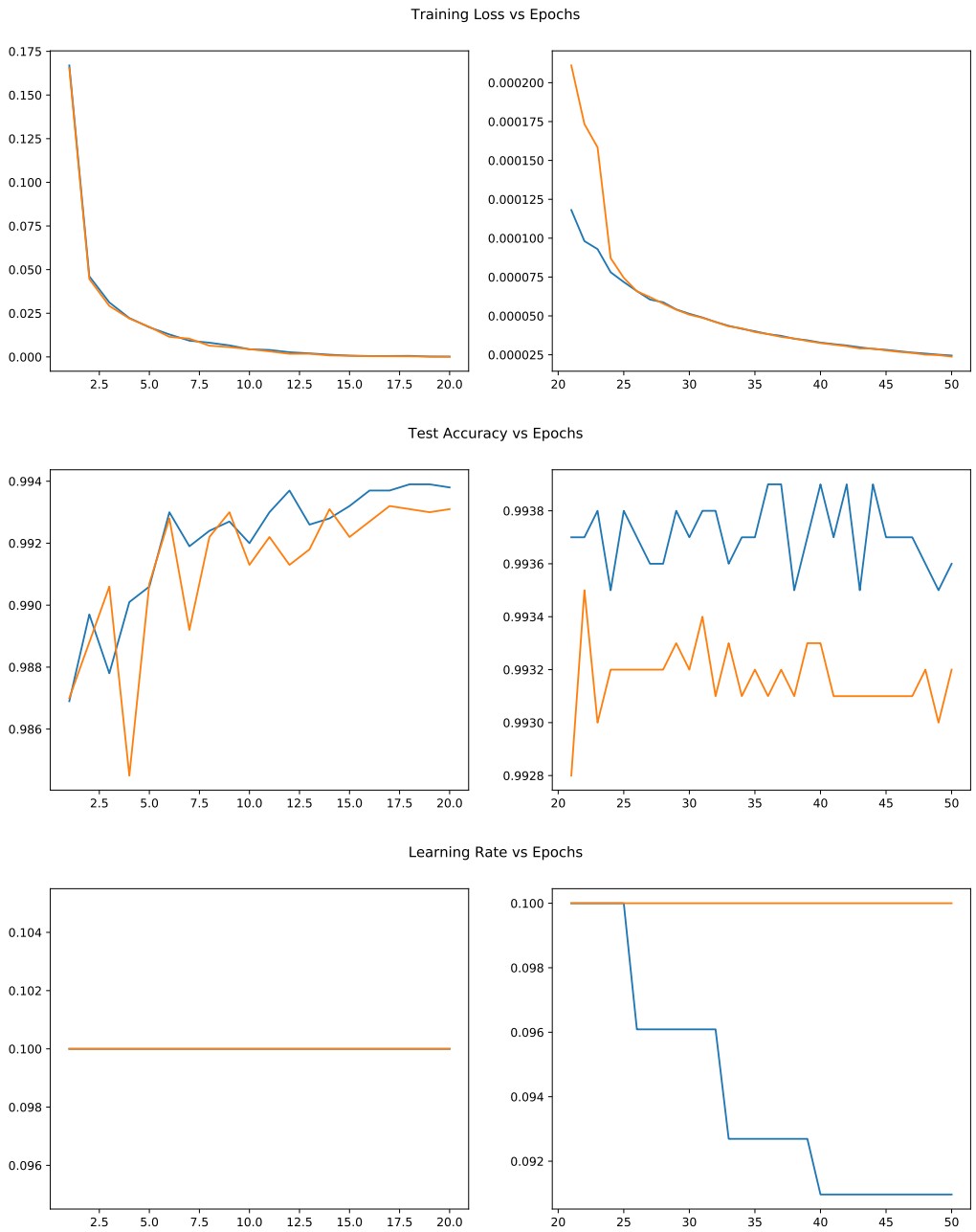

Figure 15: MNIST on Resnet-18 trained with SGD. Shown are the training loss, test accuracy and learning rate as a function of epochs, for the baseline scheme (orange) vs the *AutoLR* scheme (blue). The plot is split into 2 parts to permit higher fidelity in the y-axis range.

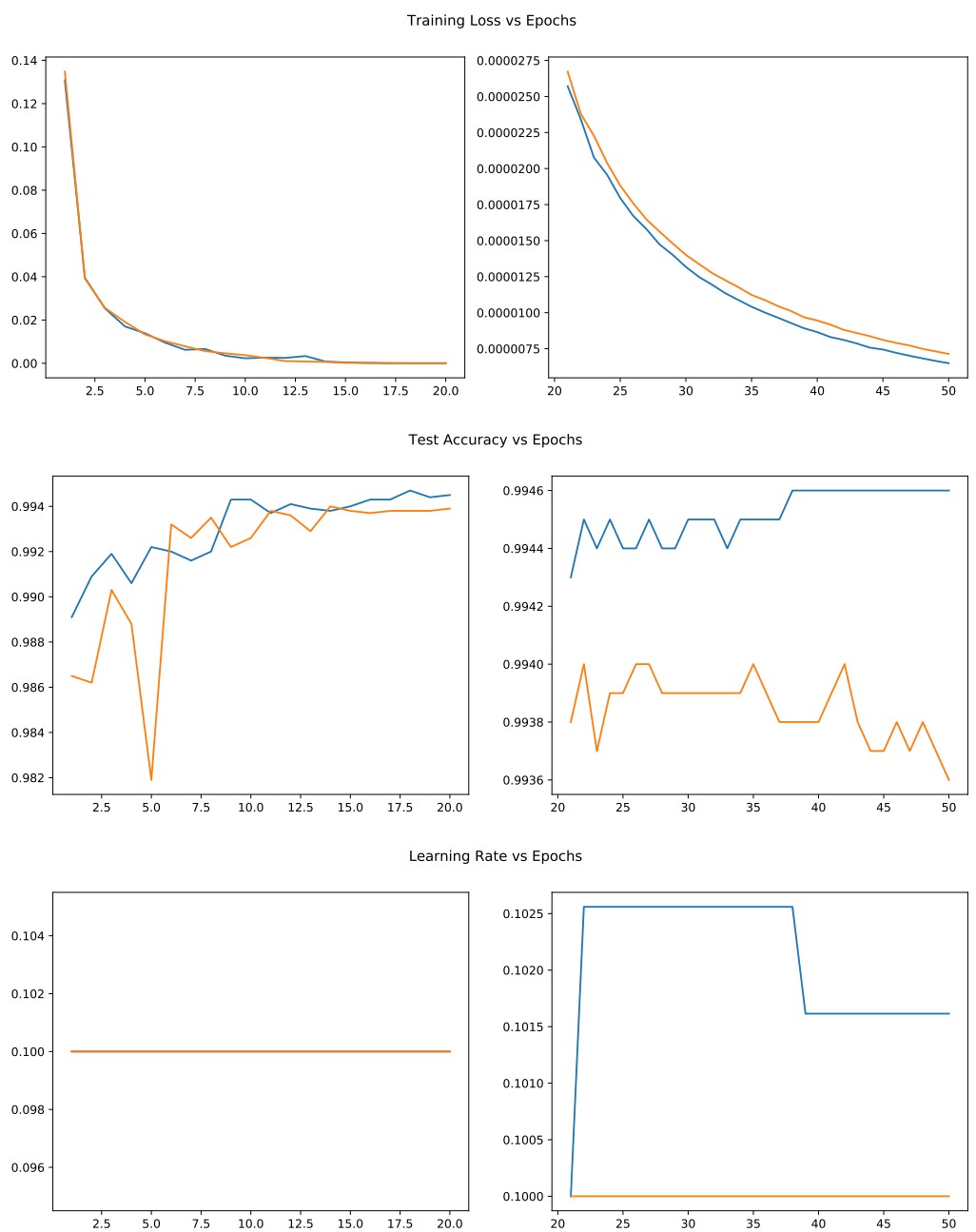

Figure 16: MNIST on Resnet-18 trained with Momentum. Shown are the training loss, test accuracy and learning rate as a function of epochs, for the baseline scheme (orange) vs the *AutoLR* scheme (blue). The plot is split into 2 parts to permit higher fidelity in the y-axis range.

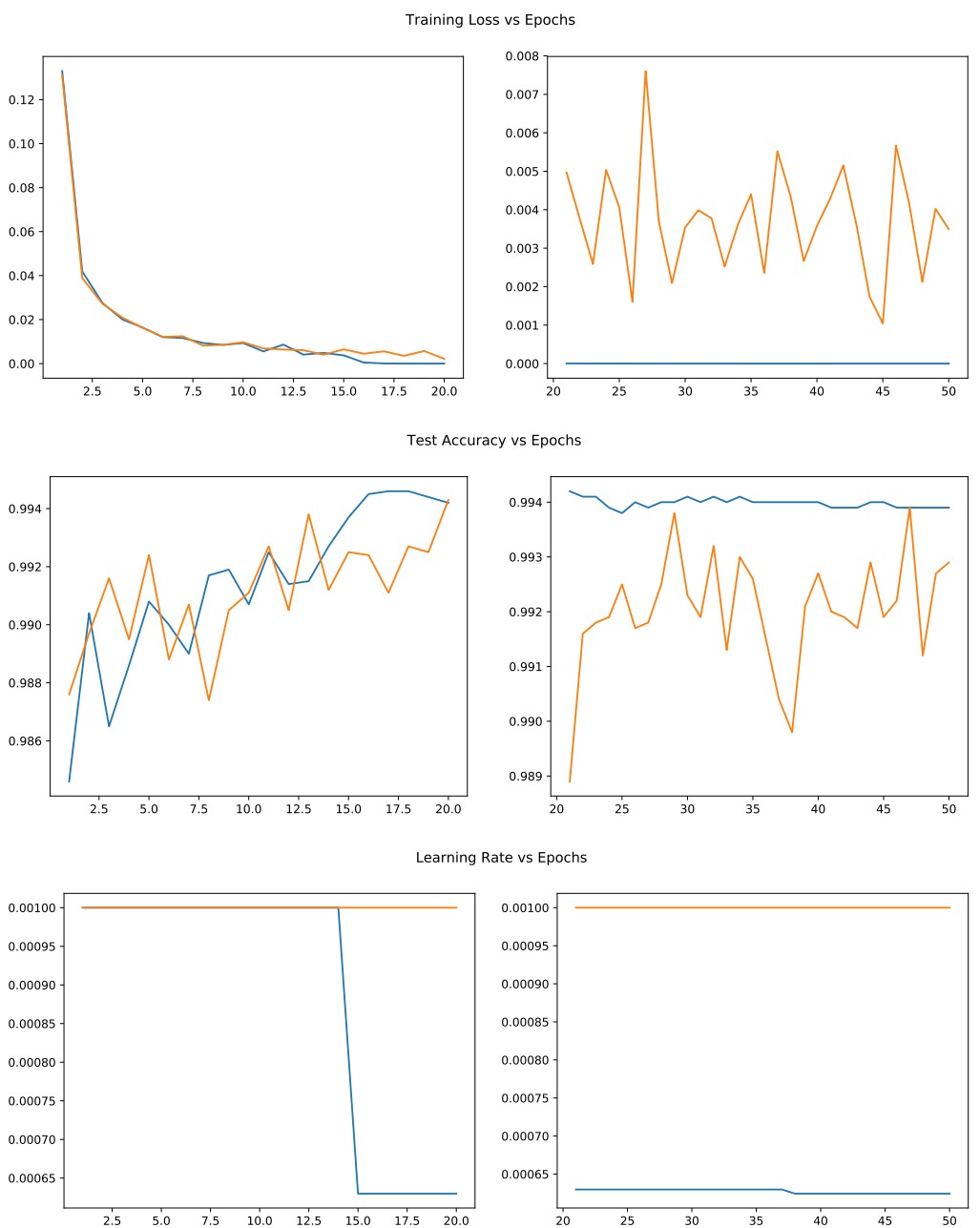

Figure 17: MNIST on Resnet-18 trained with Adam. Shown are the training loss, test accuracy and learning rate as a function of epochs, for the baseline scheme (orange) vs the *AutoLR* scheme (blue). The plot is split into 2 parts to permit higher fidelity in the y-axis range.

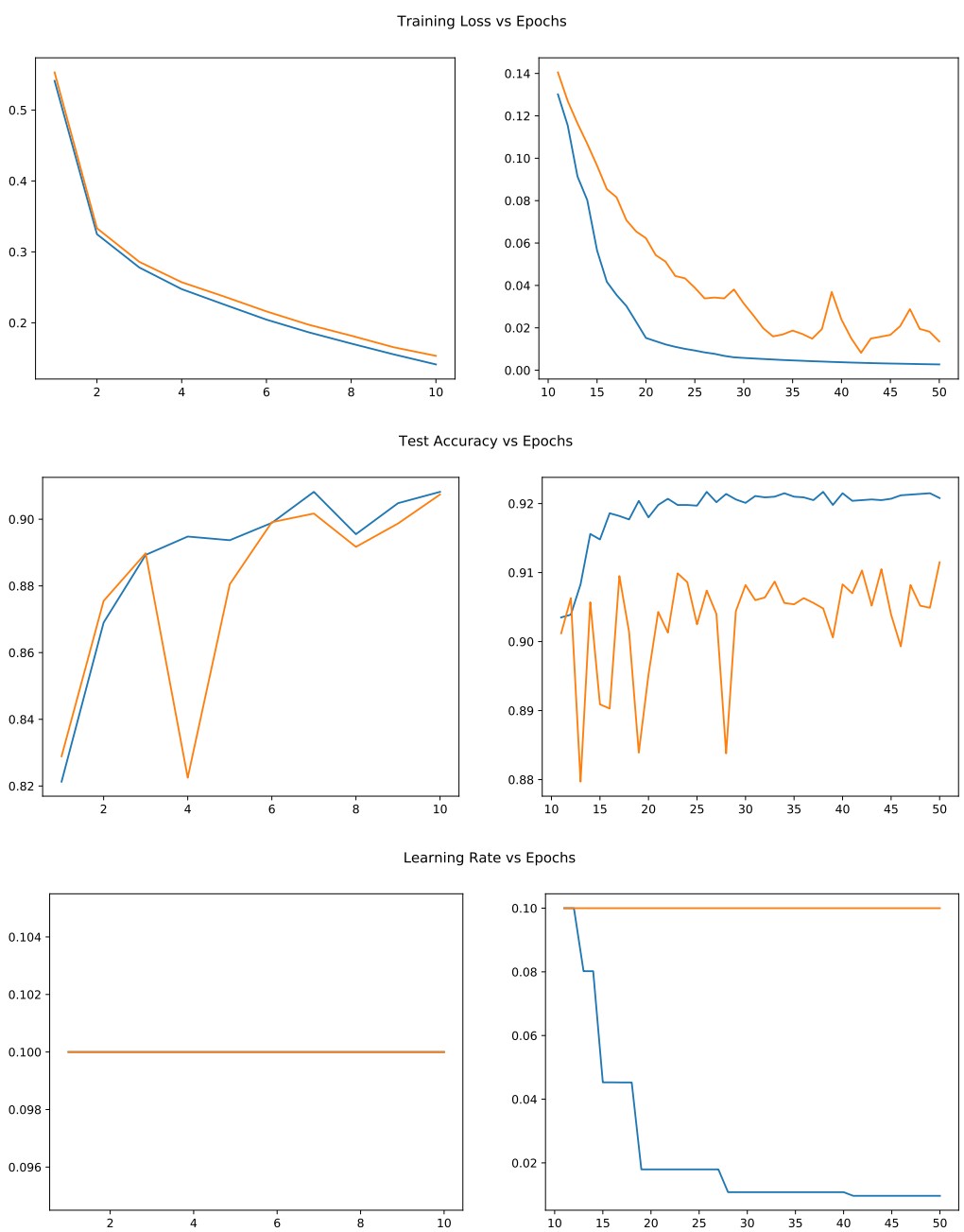

Figure 18: Fashion MNIST trained with SGD. Shown are the training loss, test accuracy and learning rate as a function of epochs, for the baseline scheme (orange) vs the *AutoLR* scheme (blue). The plot is split into 2 parts to permit higher fidelity in the y-axis range.

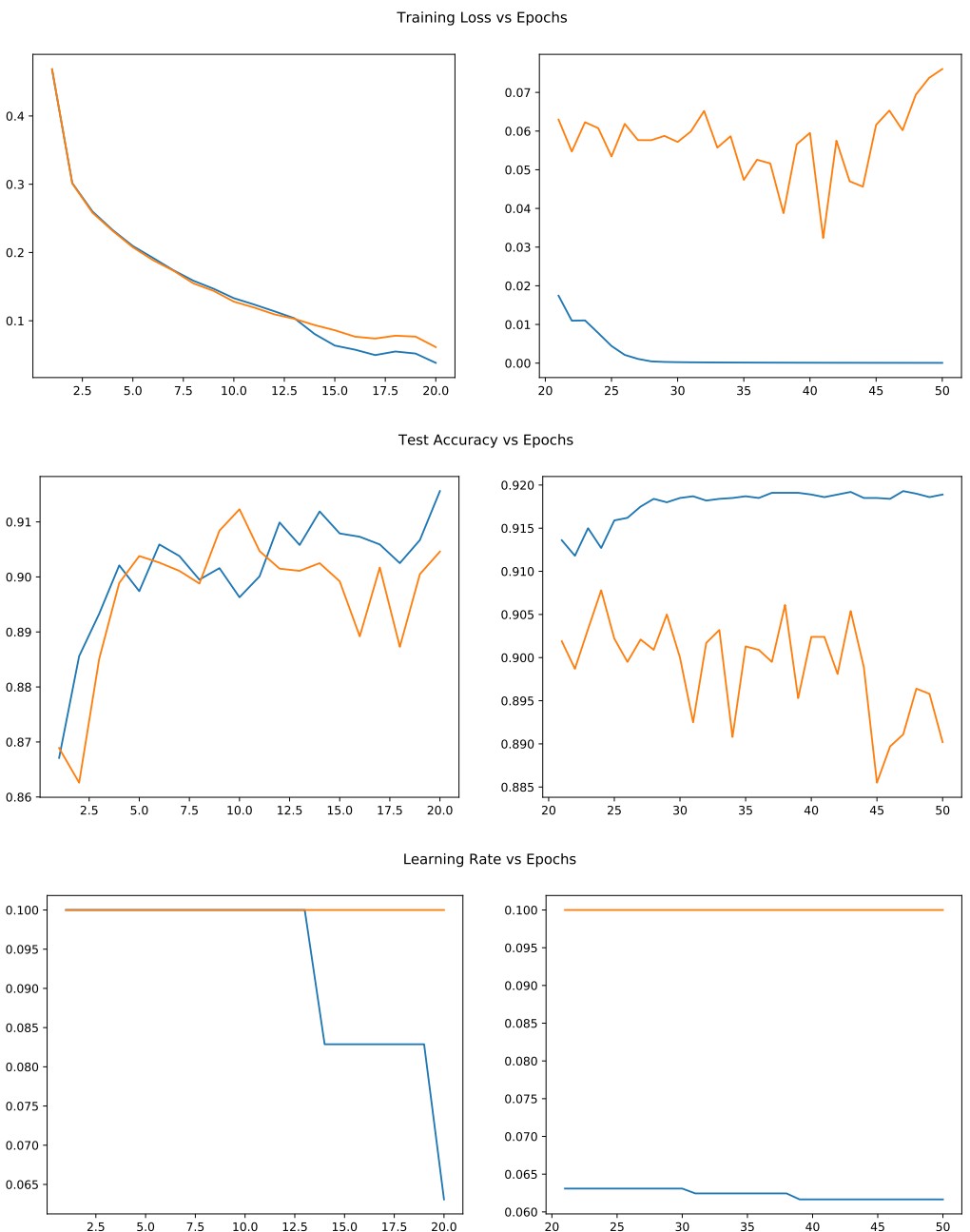

Figure 19: Fashion MNIST trained with Momentum. Shown are the training loss, test accuracy and learning rate as a function of epochs, for the baseline scheme (orange) vs the *AutoLR* scheme (blue). The plot is split into 2 parts to permit higher fidelity in the y-axis range.

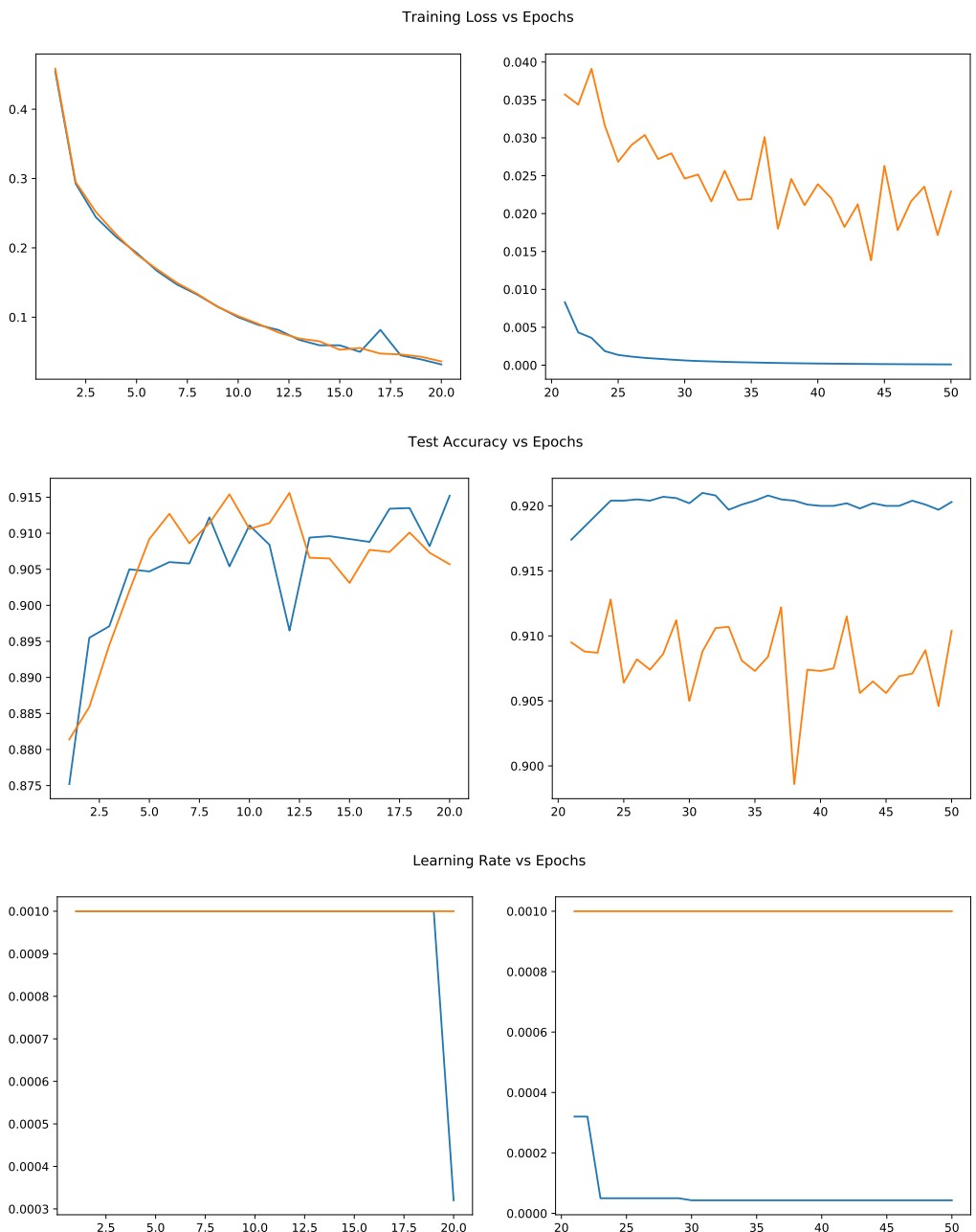

Figure 20: Fashion MNIST trained with Adam. Shown are the training loss, test accuracy and learning rate as a function of epochs, for the baseline scheme (orange) vs the *AutoLR* scheme (blue). The plot is split into 2 parts to permit higher fidelity in the y-axis range.

