# OpenReview forum: "AutoLR: A Method for Automatic Tuning of Learning Rate"
_ICLR.cc/2020/Conference — Reject_

### Official Review · AnonReviewer1 · 2019-10-22
**Official Blind Review #1**

**Rating:** 6

**Review:**

The paper proposes an automatic tuning scheme for learning rate while training neural networks. Since learning rate is the most sensitive and important hyperparameter during training, an automatic versatile method for choosing learning rate for various workloads would be of great importance.

The proposed method works for various optimizers (SGD/Momentum/Adam shown in the paper) and shown to perform as good or better than typical baseline learning rate schedule practitioners use to obtain competitive performance. There are a good number of empirical checks on both image and language tasks.

While I’m not fully familiar with literature on automatic learning rates, the authors claim that this is the first auto learning rate tuning scheme to achieve SOTA performance.

Two main components of the scheme are based on observation that initial high learning rate seemingly without making any improvements is essential for obtaining good final performance. Therefore initially there is an `explore’ phase and then in exploit phase quadratic local approximation around high learning rate update is used. Novelty in the exploit phase is expansion respect to perturbation around some potentially large step size instead of assuming step size is small.

One main concern of the proposed method is the choice of seed learning rate and duration of explore phase is still somewhat arbitrary and requires tuning for specific model/dataset. For example, in CIFAR/ResNet there are experiments showing that duration of `explore’ phase is important and choose 50 epochs. In the case of BERT for SQUAD fine tuning 2500 explore steps(half epoch) are chosen. It seems choice needs to be tuned to get good performance and with this the proposed scheme is semi-auto tuning at best. Similar points for seed learning rate could be made.

In order for the proposed method to be fully successful, either authors need to show insensitivity to general choices of seed learning rate and explore phase, or an automatic method to choose good values. Without that  I do not see significant improvement beyond tuning learning rate schedule via linear or cosine decay schemes (which is also known to perform comparatively to multiple drop schemes).

In that spirit I think an interesting baseline to compare is tuning learning rate schedule.

The proposed explore-exploit method has great potential in terms of generality and strong performance on various tasks. I would happily raise my score if the main concern is addressed, however at this point I slightly lean toward rejection.

UPDATE POST AUTHOR RESPONSE:
I thank the authors for carefully responding to the concerns raised in the initial review. I do suggest that refraining calling current method as 'auto' since it could be misleading. I think full automatic learning rates, if it can be done, will have great impact in neural network optimization. I have raised my score as author's response cleared some of issues (sensitivity to seed / comparison with other common tuned schedules).

**Experience Assessment:**

I have published one or two papers in this area.

**Review Assessment: Checking Correctness Of Derivations And Theory:**

I carefully checked the derivations and theory.

**Review Assessment: Checking Correctness Of Experiments:**

I assessed the sensibility of the experiments.

**Review Assessment: Thoroughness In Paper Reading:**

I read the paper at least twice and used my best judgement in assessing the paper.

---

> ### Author Response · Authors · 2019-11-13
> **Response to Reviewer #1 (part 1)**
>
> Thanks a lot for the review and insights. We have tried to address all concerns/questions below:
>
> (1) The main concern is regarding the tuning of hyperparameters such as seed learning rate and explore duration. Reviewer #2 had similar concerns.
>
> We agree with your observation that our method is not fully automated and does require hyperparameters. We understand that the name AutoLR can be misleading in this regard, and we will be happy to change it. We are thinking of renaming it to “AssistLR”. Please let us know if you think this name does more justice to the method and we will go ahead and update the paper with it.
>
> We would like to stress that, although our method has hyperparameters, it simplifies the problem of finding the best learning rate schedule significantly, since we now have to tune just two key hyperparameters as compared to tuning the full schedule. The full schedule is a very high dimensional function (from iteration number to learning rate), with dimension = number of iteration steps (typically of the order 100,000). Because of this high dimension, tuning for the best schedule directly is intractable, and practitioners typically rely on intuition on how different function classes (e.g. cosine, inverse square root decay, warmup, etc.) perform and try out different combinations to find the best schedule. This difficulty is also evident from Table 7 (also pasted below), which shows that for fixed function classes of learning rate schedules, it is not just important to tune the hyperparameters of each function class, but also try out different function classes. Since the space of function classes is huge in itself, searching for the best schedule is typically intractable. A case in point is our experiment of Bert finetuning with SqUAD where we were able to improve the Bert paper’s result from 80.8->81.2 by just changing the learning rate schedule. The fact that the original authors used a suboptimal learning rate points to the general difficulties in learning rate tuning. Thus, we strongly believe that even though our system is not fully automatic, it significantly simplifies the process of tuning the learning rate schedule.
>
> +----------------+-----------+------------+----------------------+----------------------+------------------------+
> |Experiment|AutoLR |Baseline |    One-Cycle       |   Cosine Decay  |  Linear Decay      |
> |                     |               |                | default | tuned | default | tuned | default | tuned |
> +----------------+-----------+------------+----------------------+----------------------+------------------------+
> |ImageNet   | 93.01    | 92.90      | -            |-            |-             |-            |-             | -          |
> +----------------+-----------+------------+----------------------+----------------------+------------------------+
> |Cifar-10       | 94.79    | 94.81      | 93.38    |94.59    |94.52    |-            |94.24     | -          |
> +----------------+-----------+------------+----------------------+----------------------+------------------------+
> |IWSLT          | 34.88    | 34.70      | 32.35    |34.57    |0.33      |34.85    |0.28       | 34.82  |
> +----------------+-----------+------------+----------------------+----------------------+------------------------+
> |SQuAD        | 81.2      | 80.7         | 79.8      |80.9      |80.8      |-            |80.7       |-           |
>
> ---- (Table 7) ----
> Caption: Test accuracies. For one-cycle we report results from the default policy of finding the max learning rate (which didn’t perform well), as well as by hand tuning it. Similarly, the default cosine decay and linear decay schedules did not work well for IWSLT, so we added a warmup phase similar to the original paper. We report results under the default and tuned columns, respectively. see section E for more details.
> ----------------------
>
> That said, we want to explore automating our scheme completely and get rid of the hyperparameters. For example, we believe that a method such as the lr range test of one-cycle can be leveraged to find the best seed learning rate for our case as well. In that respect, we did an initial seed lr sensitivity test for the Cifar-10 dataset (See Table 12. Also pasted below in point (3)), and found that our method performs well between seed lrs of 0.075 and 0.1. It turns out that the lr-range test yielded a minimum at 0.09 which is within this range. We are working on running experiments on more datasets with the lr-range test to see if we can automate this hyperparameter.
>
> Also, for the number of explore epochs hyperparameter, we want to devise a way of automatically finding if we have reached the vicinity of a wide minima; and automatically switch to exploit phase then. We unfortunately cannot use eigenvalue analysis (typical metric used for characterizing wide vs narrow minima) for this, as that characterization make sense only near the minima but not far from it (also see [3]).

---

> ### Author Response · Authors · 2019-11-13
> **Response to Reviewer #1 (part 2)**
>
> (2) “In order for the proposed method to be fully successful, either authors need to show insensitivity to general choices of seed learning rate and explore phase, or an automatic method to choose good values.”
>
> As per this suggestion, we did a sensitivity analysis of the choice of seed lr (see Table 12 also pasted below, and section F), and found that although the choice of seed lr does impact the final accuracy, AutoLR is not highly sensitive to it especially in the lower ranges. As discussed in pt (1) the lr range test of one-cycle looks promising in this regard for automating the choice of seed lr. Our method is however sensitive to the explore phase duration as shown in Table-2. As discussed in pt (1), we are exploring ways of automating this as well.
>
>  +-------------+--------------------+
> |Seed LR   | Test Accuracy |
> +-------------+--------------------+
> |0.05          | 94.60                |
> +-------------+--------------------+
> |0.075        | 94.71                |
> +-------------+--------------------+
> |0.1            | 94.79                |
> +-------------+--------------------+
> |0.125        | 94.58                |
> +-------------+--------------------+
> |0.15          | 94.25                |
> +-------------+--------------------+
>
> ----(Table-12)-----
>
> (3) “Without that  I do not see significant improvement beyond tuning learning rate schedule via linear or cosine decay schemes (which is also known to perform comparatively to multiple drop schemes).”
>
> To check this we have run all our examples (except ImageNet which is currently running) with linear and cosine decay schemes (as well as one-cycle learning rate suggested by Reviewer #2), and have updated the paper with those results. and have updated the paper with those results. Please see Tables 9, 10, 11 for detailed numbers. Table 7 (also pasted in pt 1) has test accuracy for our method and all other baseline schedules we tried. As shown, AutoLR compares favorably against all other learning rate schedules, except slightly missing out in Cifar-10. An interesting observation is that both cosine and linear decay perform better than the inverse square root decay learning rate schedule suggested in the “Attention is all you need” paper. A similar observation can be made for the Bert fine-tuning for SQuAD. The fact that the original authors used a suboptimal learning rate points to the general difficulties in learning rate tuning, and how an automated method can be useful in alleviating this problem.
>
>
> (4) “In that spirit I think an interesting baseline to compare is tuning learning rate schedule.”
>
> We did not understand what you mean by the baseline of tuning learning rate schedule. Please can you elaborate or give references for this schedule and we will run experiments with it. If you meant general tuning of the learning rate schedule, as mentioned above, this is very hard and intractable because of the very high dimensionality of learning rates schedules. E.g. in Cifar-10 on ResNet example, the dimensionality is 78200. However, we can try more function classes such as linear, cosine decay, etc. Please let us know if you would like us to run experiments with more function classes.
>
>
> [1] Leslie N Smith, Cyclical Learning Rates for Training Neural Networks, https://arxiv.org/pdf/1506.01186.pdf
>
> [2] Leslie N Smith, A disciplined approach to neural network hyper-parameters: Part 1 -- learning rate, batch size, momentum, and weight decay, https://arxiv.org/pdf/1803.09820.pdf
>
> [3] Jastrzebski et al, On the Relation Between the Sharpest Directions of DNN Loss and the SGD Step Length, https://arxiv.org/abs/1807.05031

---

### Official Review · AnonReviewer2 · 2019-10-22
**Official Blind Review #2**

**Rating:** 3

**Review:**


The authors propose an automatic learning rate schedule based on an explore (always increase LR initially) and then exploit (more typical patience based decay) strategy. The strategy seems to factor in recent understanding of deep learning optimization, and I was very much convinced by the overall idea, even if not fully convinced by the motivation. The main issues with the paper I have is that (1) I does not compare to a strong enough baseline (one cycle), and (2) the schedule does not feel automatic enough to call it "automatic"; see detailed comments. Finally, I found the remarks and analysis regarding width of the minima, and the stipulation that there are more narrow minimas, not substantiated and even contradicatory to some of the results in the literature. Based on this, I am currently leaning towards rejecting the paper. I am willing to raise my score if issues with the experimental setting are addressed.

Detailed comments:

1. I am not convinced that the baseline is strong enough.

While I appreciate the extensive range of architecture and datasets, I am not convinced that the baseline schedule tested against in each setting is actually "state of the art" as stated in the abstract. To the best of my knowledge, the state-of-the-art learning rate schedule, which uses a similar computational budget as the proposed method, is one cycle. In particular, one cycle uses LR range test to select the appropriate starting learning rate and then warmups to it gradually; see [1] for more details (see also [2]).

Note that one cycle includes warming up the learning rate, which is another motivation for including it. As seen for instance in Fig. 4 the proposed AutoLR can increase learning rate initially, while the hand tuned schedules don't. This does not seem to be a fair comparison, given that the state of the art schedules in vision models very often do warm-up the learning rate.

2. The learning rate schedule requires feeding in "seed" learning rate. Is it an automatic learning rate schedule then? I am a bit confused by this claim. Could you please clarify what is the claim about the method?

3. To sum up point 1 and 2, while I like what the paper set out to do, I think that it is key to either (1) demonstrate that the automatic schedule is substantially better than the state of the art schedule, or (2) demonstrate that initial seed learning rate is not an important hyperparameter. If both are not true, then how would you convince the reader to use the method in practice?

4. It is implicitely assumed in the analysis that wide minima are good for generalization. It seems to me that recent evidence points towards the direction that width of the minima is a epiphenomenon, see [3,4,5,6] experiments. I think at least a discussion of (some) of these papers is very important. Also it is worth noting that [5] studied how a high learning rate (or a small batch size) enters wide regions of the loss surface early.

5. "In that respect, an interesting intuitive observation is that a large learning rate can escape narrow minima “valleys” easily (as the optimizer can jump out of them with large steps), however once it reaches a wide minima “valley”, it is likely to get stuck in it (if the “width” of the wide valley is large compared to the step size).". I think [5,8] should be cited here. Both papers studied how learning rate selects minima shape.

6. At the same time, I am not convinced by the experimental data for "We hypothesize that for deep neural networks, narrow minimas far outnumber the wide minimas.":

6a) Taking on the face value that the experiment shows that a low learnring rate can find a wide minima, this doesn't prove the hypothesis. At best (i.e. assuming the indeed there is a strong correlation between width of the low lr minima, and generalization) it proves that low learning rate can find wide minima. It doesn't establish any bound/estimation on the relative number of narrow to wide minima. What if low learning rate by default is driven to narrow minima, and high learning rate is driven to wide minima, but the two are equal in number? I think a similar experimental data would be observed in such a null world.

6b) More importantly, it is not demonstrated that the one experiment that worked better had lower curvature. It seems implicitely assumed that if it worked better then it for sure ended up in a wider minima. I do not think this is a valid reasoning given the weak data for a causal relationship between curvature and generalization (see 2). Could you please report curvature explicitely, rather than making this assumption?

References:

[1] Fast.ai documentation on one cycle method, https://docs.fast.ai/callbacks.one_cycle.html
[2] Arpit et al, Walk with SGD, https://arxiv.org/abs/1802.08770
[3] Golatkar et al, Time Matters in Regularizing Deep Networks: Weight Decay and Data Augmentation Affect Early Learning Dynamics, Matter Little Near Convergence, https://arxiv.org/abs/1905.13277
[4] Yoshida et al, Spectral Norm Regularization for Improving the Generalizability of Deep Learning, https://arxiv.org/pdf/1705.10941.pdf
[5] Jastrzebski et al, On the Relation Between the Sharpest Directions of DNN Loss and the SGD Step Length, https://arxiv.org/abs/1807.05031
[6] Guiroy et al, Towards Understanding Generalization in Gradient-Based Meta-Learning, https://arxiv.org/abs/1907.07287
[7] Wang et al, Identifying generalization properties in neural networks, https://arxiv.org/pdf/1809.07402.pdf
[8] Wu et al, How SGD Selects the Global Minima in Over-parameterized Learning: A Dynamical Stability Perspective, https://papers.nips.cc/paper/8049-how-sgd-selects-the-global-minima-in-over-parameterized-learning-a-dynamical-stability-perspective.pdf



**Experience Assessment:**

I have published one or two papers in this area.

**Review Assessment: Checking Correctness Of Derivations And Theory:**

N/A

**Review Assessment: Checking Correctness Of Experiments:**

I assessed the sensibility of the experiments.

**Review Assessment: Thoroughness In Paper Reading:**

I read the paper thoroughly.

---

> ### Author Response · Authors · 2019-11-13
> **Response to Reviewer #2 (part 1)**
>
> Thanks a lot for the review and insights. We have tried to address all concerns/questions below:
>
> - (1) Baselines:
>
> For recent models, we used the baseline schedules which were specified in the corresponding papers. E.g. in IWSLT on Transformers, we used the schedule mentioned in the “Attention is all you need” paper [1], and in Bert finetuning for SQuAD we used the schedule mentioned in the Bert paper [2]. For Cifar and ImageNet on ResNet we tried to find the best baselines used by various papers and opensource implementations.
>
> We were not aware of the one-cycle policy and thanks for bringing it to our attention. We have run all our examples (except ImageNet which is currently running) with the one-cycle baseline (as well as linear and cosine decay schemes suggested by Reviewer #1), and have updated the paper with those results. Please see Tables 9, 10, 11 for detailed numbers. Also see Section E for details on the lr range test for the one-cycle runs. Table 7 (also pasted here) has test accuracy for our method and all other baseline schedules we tried. For one-cycle we report results from the default policy of finding the maximum learning rate (which didn’t perform very well), as well as by hand tuning it (see section E for more details). Similarly, the default cosine decay and linear decay schedules did not work well for IWSLT, so we added a warmup phase in the beginning as is done in the original paper’s baseline. We report both these results under the default and tuned columns, respectively. As shown, AutoLR compares favorably against all other learning rate schedules, except slightly missing out in Cifar-10. An interesting observation is that both cosine and linear decay perform better than the inverse square root decay learning rate schedule suggested in the “Attention is all you need” paper. A similar observation can be made for the Bert fine-tuning for SQuAD.
>
> +----------------+-----------+------------+----------------------+----------------------+------------------------+
> |Experiment|AutoLR |Baseline |    One-Cycle       |   Cosine Decay  |  Linear Decay      |
> |                     |               |                | default | tuned | default | tuned | default | tuned |
> +----------------+-----------+------------+----------------------+----------------------+------------------------+
> |ImageNet   | 93.01    | 92.90      | -            |-            |-             |-            |-             | -          |
> +----------------+-----------+------------+----------------------+----------------------+------------------------+
> |Cifar-10       | 94.79    | 94.81      | 93.38    |94.59    |94.52    |-            |94.24     | -          |
> +----------------+-----------+------------+----------------------+----------------------+------------------------+
> |IWSLT          | 34.88    | 34.70      | 32.35    |34.57    |0.33      |34.85    |0.28       | 34.82  |
> +----------------+-----------+------------+----------------------+----------------------+------------------------+
> |SQuAD        | 81.2      | 80.7         | 79.8      |80.9      |80.8      |-            |80.7       |-           |
> +----------------+-----------+------------+----------------------+----------------------+------------------------+
>
> ---- (Table 7) ----

---

> > ### Comment · AnonReviewer2 · 2019-11-13
> > **Thank you for the rebutal**
> >
> > I really appreciate the additional experimental data and the updates to the paper, but unfortunately I would like to keep my score. I am keeping my score on the following grounds:
> >
> > (1) I find it difficult to believe one cycle (or warming up) learning rate doesn't help (or just 0.02) compared to a baseline. To the best of my knowledge it is a very common practice to do so, and part of many state of the art results,
> >
> > (2) The AutoLR method is actually quite sensitive to the base learning rate, which makes it muddied what is the contribution. It seems to me the paper would benefit from a rewrite that really clarifies the contribution (changing the name might be a good starting point).
> >
> > Also, thank you for the changes in the hypothesis section. I do agree that length of training is important, and it is not clear why it is important. One additional point - I think you should also mention the training loss. The top eigenvalues of the Hessian are, mathematically speaking,. expected to shrink when training loss is lower (if cross entropy is used). This is mentioned also in Wu et al. Hence, the effect of longer training on Hessian might be simply due to reducing loss more.

---

> > > ### Author Response · Authors · 2019-11-14
> > > **Response to Reviewer #2 comments**
> > >
> > > Thanks a lot for your response.
> > >
> > >
> > > Q: (1) I find it difficult to believe one cycle (or warming up) learning rate doesn't help (or just 0.02) compared to a baseline. To the best of my knowledge it is a very common practice to do so, and part of many state of the art results,
> > >
> > > Response:
> > >
> > > - Please can you point us to references of SOTA accuracy results with one-cycle, as we were not able to find any. We have seen one-cycle being used for DAWN style benchmarks which compete on time to accuracy, but not on final accuracy.
> > >
> > > - We would like to re-emphasize that the baselines we used are typically highly hand tuned by authors or practitioners and are thus typically very hard to beat. Thus, it is not very surprising that one-cycle is not able to match it in most examples.
> > >
> > > - For your convenience, we have uploaded the cifar code here: https://drive.google.com/drive/folders/1MftbzAiXpAB3EKvSdr9xkP_lNAR9MU1h?usp=sharing, with instructions in README to reproduce the results with both custom step schedule baseline and one-cycle. Note that we found one-cycle to be very sensitive to maximum learning rate choice, and have mentioned in README the value which performed well in our experiments.
> > >
> > > - Also we are not sure what comparison are you pointing to when you say one-cycle helps by just 0.02. One-cycle only performed better than the default baselines in only the SQuAD fine-tuning experiment by 0.2 and underperformed in others.
> > >
> > > ---------------
> > >
> > > Q: One additional point - I think you should also mention the training loss. The top eigenvalues of the Hessian are, mathematically speaking,. expected to shrink when training loss is lower (if cross entropy is used). This is mentioned also in Wu et al. Hence, the effect of longer training on Hessian might be simply due to reducing loss more.
> > >
> > > Response:
> > >
> > > We have updated the paper with training loss numbers, and they are all very similar, as can be seen in table below:
> > >
> > > +--------------+-----------------+--------------------+
> > > |Accuracy  |  Eigenvalue |   Training loss |
> > > +--------------+-----------------+--------------------+
> > > |94.98         | 0.02              | 0.00152            |
> > > +--------------+-----------------+--------------------+
> > > |94.58         | 0.05              | 0.00156            |
> > > +--------------+-----------------+--------------------+
> > > |94.32         | 0.10              | 0.00150            |
> > > +--------------+-----------------+--------------------+
> > >
> > > Also, just to clarify all the experiments were trained with the same learning rate schedule and the same number of iterations, so there should be no effect of longer training.

---

> > > > ### Comment · AnonReviewer2 · 2019-11-14
> > > > **Clarification**
> > > >
> > > > What I meant by " I find it difficult to believe one cycle (or warming up)" is that I find it surprising if neither one cycle, nor warm-up + keep constant (which is actually more standard than one-cycle), would underperform baseline. To check this it would be necessary to add warm-up to the table. I think that would make the experimental results much more convincing.
> > > >
> > > > Having said that, there is indeed a difference between DAWNBench, and your setting. One paper I am aware of that compares learning rate schedules in a more related setting to yours is https://arxiv.org/abs/1802.08770. And there it is visible that one-cycle and warm-up beat the baseline. Arguably, this conclusion might not carry over to your setting, but a comparison seems necessary given the truly wide-spread use of warm-up.
> > > >
> > > > Thank you for your clarification on the eigenvalues. It seems that more data would be needed to establish this correspondence (but I do not hold this as a weakness. I understand you put this as a hypothesis in the paper).

---

> > > > > ### Author Response · Authors · 2019-11-15
> > > > > **Request for clarification on experiments to run**
> > > > >
> > > > > Thanks a lot for your response. We are planning to start warmup experiments right away so that we can post results before the response window closes today. We had a few clarifications regarding that and will really appreciate if you can help us with those soon.
> > > > >
> > > > > We went over the https://arxiv.org/abs/1802.08770 (A Walk with SGD) paper. There they don’t seem to have a warm-up schedule but a trapezoidal schedule (see section C and figure 8) -- which is essentially linear warmup + constant + linear decay. Please can you clarify if you meant trapezoidal when you said warmup. We are planning to start runs with trapezoidal right away. We will also run just warmup + constant runs but in our experience these may not give good results.
> > > > >
> > > > > Thanks a lot.

---

> ### Author Response · Authors · 2019-11-13
> **Response to Reviewer #2 (part 2)**
>
> - (2) How automatic is AutoLR:
>
> In the paper and title we had introduced our method as a method for automatic “tuning” of learning rate. It does start with a given seed lr and then tries to tune the schedule. However, as correctly pointed out by both Reviewer #1 and #2, our method is not fully automated and does require hyperparameters. We understand that the name AutoLR can be misleading in this regard, and we will be happy to change it. We are thinking of renaming it to “AssistLR”. Please let us know if you think this name does more justice to the method and we will go ahead and update the paper with it.
>
> We would like to stress that, although our method has hyperparameters, it simplifies the problem of finding the best learning rate schedule significantly, since we now have to tune just two key hyperparameters as compared to tuning the full schedule. The full schedule is a very high dimensional function (from iteration number to learning rate), with dimension = number of iteration steps (typically of the order 100,000). Because of this high dimension, tuning for the best schedule directly is intractable, and practitioners typically rely on intuition on how different function classes (e.g. cosine, inverse square root decay, warmup, etc.) perform and try out different combinations to find the best schedule. This difficulty is also evident from Table 7 (pasted above in point 1), which shows that for fixed function classes of learning rate schedules, it is not just important to tune the hyperparameters of each function class, but also try out different function classes. Since the space of function classes is huge in itself, searching for the best schedule is typically intractable. A case in point is our experiment of Bert finetuning with SqUAD where we were able to improve the Bert paper’s result from 80.8->81.2 by just changing the learning rate schedule. The fact that the original authors used a suboptimal learning rate points to the general difficulties in learning rate tuning. Thus, we strongly believe that even though our system is not fully automatic, it significantly simplifies the process of tuning the learning rate schedule.
>
> That said, we want to explore automating our scheme completely and get rid of the hyperparameters. For example, we believe that a method such as the lr range test of one-cycle can be leveraged to find the best seed learning rate for our case as well. In that respect, we did an initial seed lr sensitivity test for the Cifar-10 dataset (See Table 12. Also pasted below in pt 3), and found that our method performs well between seed lrs of 0.075 and 0.1. It turns out that the lr-range test yielded a minimum at 0.09 which is within this range. We are working on running experiments on more datasets with the lr-range test to see if we can automate this hyperparameter.
>
> Also, for the number of explore epochs hyperparameter, we want to devise a way of automatically finding if we have reached the vicinity of a wide minima; and automatically switch to exploit phase then. We unfortunately cannot use eigenvalue analysis (typical metric used for characterizing wide vs narrow minima) for this, as that characterization make sense only near the minima but not far from it (also see [3]).
>
> - (3)
>
> We have compared our method with more baselines (one-cycle, cosine and linear decay), and as shown in Table 7, AutoLR performs favorably in all (only slightly missing out in Cifar), including providing non-trivial gains in Bert fine-tuning for SQuAD.
>
> We did a sensitivity analysis of the choice of seed lr (see section F and Table 12 also pasted below), and found that although the choice of seed lr does impact the final accuracy, AutoLR is not highly sensitive to it especially in the lower ranges.
>
> +-------------+--------------------+
> |Seed LR   | Test Accuracy|
> +-------------+--------------------+
> |0.05          | 94.60                |
> +-------------+--------------------+
> |0.075        | 94.71                |
> +-------------+--------------------+
> |0.1            | 94.79                |
> +-------------+--------------------+
> |0.125        | 94.58                |
> +-------------+--------------------+
> |0.15          | 94.25                |
> +-------------+--------------------+
>
> -----(Table 12)-----
>
> - (4) Wide minima and generalization.
>
> Thanks for pointing us to these references. We have added a discussion of these references in the paper. We agree that there is not a complete consensus on the relation between wide minima and generalization, but in our understanding from reading many results in this area, it seems that in at least the common case, wide minima and generalization are typically correlated.
>
> - (5) “I think [5,8] should be cited here. Both papers studied how learning rate selects minima shape.”
>
> We have updated the paper to cite these two references here. We had cited the first reference already but in a different context.

---

> ### Author Response · Authors · 2019-11-13
> **Response to Reviewer #2 (part 3)**
>
> -(6) I am not convinced by the experimental data for "We hypothesize that for deep neural networks, narrow minimas far outnumber the wide minimas."
>
> - We have now added more discussion of the hypothesis in the paper (explaining our hypothesis better in Section 2.1 and discussion of related work that adds more evidence to this hypothesis). We agree that evidence for this hypothesis is still limited and further investigation is required to fully validate it, so we have removed any reference to validation of hypothesis.
>
> - (6a)
>
> > Q: “Taking on the face value that the experiment shows that a low learning rate can find a wide minima, this doesn't prove the hypothesis. At best (i.e. assuming the indeed there is a strong correlation between width of the low lr minima, and generalization) it proves that low learning rate can find wide minima.”
>
> A: We are assuming that you meant high learning rate instead of low learning rate finds wide minima in the above sentence. Please correct us if we are wrong.
>
> > Q: “What if low learning rate by default is driven to narrow minima, and high learning rate is driven to wide minima, but the two are equal in number? I think a similar experimental data would be observed in such a null world.”
>
> A:  Although it could likely be the case that high learning rate drives to a wide minima  (and low learning rate to narrow minima), this by itself does not fully explain the experimental observations, specifically,  the observation that *training at high learning rate for long duration consistently achieves high test accuracy while training at high learning rate for shorter duration until train loss saturates rarely achieves high test accuracy*. Consider the twenty runs of training on CIFAR10 that use a high learning rate for 40 epochs and then the rate is decayed. We find that only one run results in high accuracy. As per your counterfactual (high learning rate always drives to wide minima), all runs in this experiment should give similar/high accuracies. Note that, if we perform twenty training runs that use the same high learning rate for 100 epochs and then decay the rate, all twenty runs do result in similar/high accuracies. Our hypothesis could be one possible explanation for this phenomena based on the following intuition:  if the density of wider minima were low, using a high learning rate for long enough duration will always find it (as high learning rate enables the optimization to jump out of the many narrow minima until it finds a wide minima and gets stuck there) but using a high learning for a short duration will only find the wide minima with some probability (based on the chance of landing in the wide minima early during training). Since we find empirically that the probability of landing in the wide minima early in training is low (about 1/20 based on our experiments), we hypothesize that wider minima are fewer in number than narrow minima. We have updated the paper to explain our hypothesis more clearly and also added discussion of an experiment from the literature that adds evidence to this hypothesis.
>
> - (6b) “it is not demonstrated that the one experiment that worked better had lower curvature. It seems implicitely assumed that if it worked better then it for sure ended up in a wider minima”
>
> Yes, we did make this implicit assumption that higher accuracy was due to wider minima. Thanks for pointing this out. We have now computed the curvature of the loss surface at the end of training for the various runs and are able to validate this assumption. Specifically, we use the highest eigenvalues of the Hessian at the minima as a metric of the minima width (see [3,4]).  We find that the high accuracy runs consistently have a smaller eigenvalues compared to low accuracy runs. For example the run with highest accuracy of 94.98 had an eigenvalue of 0.02, while the run with median accuracy of 94.58 had an eigenvalue of 0.05, and the run with minimum accuracy of 94.32 had an eigenvalue of 0.10. We have updated the paper with these numbers.
>
>
> [1] Vaswani et al., Attention is All you Need, https://papers.nips.cc/paper/7181-attention-is-all-you-need.pdf
>
> [2] Devlin et al., BERT: Pre-training of Deep Bidirectional Transformers for Language Understanding, https://arxiv.org/abs/1810.04805
>
> [3] Jastrzebski et al., On the Relation Between the Sharpest Directions of DNN Loss and the SGD Step Length, https://arxiv.org/abs/1807.05031
>
> [4] Keskar et al., On Large-Batch Training for Deep Learning: Generalization Gap and Sharp Minima, https://arxiv.org/abs/1609.04836

---

### Author Response · Authors · 2019-11-15
**Comparison results with learning rate schedules like one-cycle, cosine-decay, linear-decay and trapezoid**


As recommended by the reviewers, we have updated the paper with comparison results of AutoLR with other learning rate schedules in the literature such as one-cycle, cosine-decay, linear-decay, and trapezoid [1] (See Table 7 in paper. Also pasted below). As reviewer#2 had pointed out and as mentioned in [1], Trapezoid which does linear warmup followed by constant lr (and finally linear decay) does perform better than the one-cycle learning rate on Cifar-10 (and IWSLT). However, it *significantly underperforms* the baseline and AutoLR in the SQuAD benchmark.

Based on the below table, we conclude that,  except AutoLR,  none of the schemes in the literature and suggested by the reviewers (one-cycle, cosine-decay, linear-decay, and trapezoid, with and without tuning) match or exceed the highly tuned baselines across all three datasets.

+----------------+-----------+------------+----------------+----------------------+-----------------------+----------------------+
|Experiment|AutoLR |Baseline | Trapezoid  |    One-Cycle       |   Cosine Decay   |  Linear Decay    |
+----------------+-----------+------------+----------------+----------------------+-----------------------+----------------------+
|                     |               |               |                     | default | tuned | default | tuned | default | tuned |
+----------------+-----------+------------+----------------+----------------------+-----------------------+----------------------+
|ImageNet   | 93.01    | 92.90      | -                  |-             |-            |-             |-            |-             | -          |
+----------------+-----------+------------+----------------+----------------------+-----------------------+----------------------+
|Cifar-10       | 94.79    | 94.81      | 94.69          | 93.38    |94.59    |94.52    |-            |94.24     | -          |
+----------------+-----------+------------+----------------+----------------------+-----------------------+----------------------+
|IWSLT          | 34.88    | 34.70      | 34.85          |32.35    |34.57    |0.33      |34.85    |0.28       | 34.82   |
+----------------+-----------+------------+----------------+----------------------+-----------------------+----------------------+
|SQuAD        | 81.2      | 80.7         | 80.1            |79.8      |80.9      |80.8      |-            |80.7       |-            |
+----------------+-----------+------------+----------------+----------------------+-----------------------+----------------------+
 Imagenet training still running.

[1] Arpit et al, Walk with SGD, https://arxiv.org/abs/1802.08770

---

> ### Comment · AnonReviewer2 · 2019-11-15
> **Thank you for results**
>
> Thank you for results. I am confused by one thing. Shouldn't Trapezoid include as a special case baseline (at least on SQuAD and IWSLT where we do not have step-wise schedule)? Could you please (1) share all the hyperparameter ranges, (2) report final training loss in the table?

---

> > ### Author Response · Authors · 2019-11-15
> > **Clarification**
> >
> > Hello,
> >
> > I am not sure what you mean by "Shouldn't Trapezoid include as a special case baseline". Yes, IWSLT and SQuAD baselines do not use step schedule --  IWSLT baseline uses warmup and inverse square root decay while SQuAD uses linear decay as a baseline. For Trapezoid for SQuAD and IWSLT,  we use the maxlr obtained for the tuned one-cycle scheme as the constant lr and use linear warmup and linear decay at the beginning and end.
> >
> > All hyper-parameters and training loss information is listed in appendix D and E in the revised version of the paper. For example, Table 9,10,11 has the final training loss/train ppl for all the schemes evaluated for the three datasets.
> >
> > Please do let us know if you have any further questions.

---

### Decision · Program_Chairs · 2019-12-19

**Decision:**

Reject

**Comment:**

The authors propose a method for automatic tuning of learning rates. The reviewers liked the idea but felt that there are much more extensive experiments to be done especially better baselines. Also, clarifying what aspect is automated is important, because no method can be truly automatic: they all have some hyperparameters.